# Unlearning in Diffusion models under Data Constraints: A Variational Inference Approach

**Subhodip Panda**                                        *subhodipp@iisc.ac.in*
*Department of ECE*
*Indian Institute of Science*

**Varun M S**                                             *varun80042@gmail.com*
*Department of Computer Science*
*PES University*

**Shreyans Jain**                                         *shreyans.jain@iitgn.ac.in*
*Department of EE*
*Indian Institute of Technology, Gandhinagar*

**Sarthak Kumar Maharana**                                *skm200005@utdallas.edu*
*Department of Computer Science*
*The University of Texas at Dallas*

**Prathosh A.P**[1,2].                                    *prathosh@iisc.ac.in*
[1] *Department of ECE, Indian Institute of Science*
[2] *LatentForce.ai*

**Reviewed on OpenReview:** *https://openreview.net/forum?id=mAHRgieyOV*

## Abstract

For a responsible and safe deployment of diffusion models in various domains, regulating the generated outputs from these models is desirable because such models could generate undesired, violent, and obscene outputs. To tackle this problem, recent works use *machine unlearning* methodology to forget training data points containing these undesired features from pre-trained generative models. However, these methods proved to be ineffective in data-constrained settings where the whole training dataset is inaccessible. Thus, the principal objective of this work is to propose a machine unlearning methodology that can prevent the generation of outputs containing undesired features from a pre-trained diffusion model in such a data-constrained setting. Our proposed method, termed as Variational Diffusion Unlearning (**VDU**), is a computationally efficient method that only requires access to a subset of training data containing undesired features. Our approach is inspired by the variational inference framework with the objective of minimizing a loss function consisting of two terms: *plasticity inducer* and *stability regularizer*. *Plasticity inducer* reduces the log-likelihood of the undesired training data points, while the *stability regularizer*, essential for preventing loss of image generation quality, regularizes the model in parameter space. We validate the effectiveness of our method through comprehensive experiments for both class unlearning and feature unlearning. For class unlearning, we unlearn some user-identified classes from MNIST, CIFAR-10, and tinyImageNet datasets from a pre-trained unconditional denoising diffusion probabilistic model (DDPM). Similarly, for feature unlearning, we unlearn the generation of certain high-level features from a pre-trained Stable Diffusion model trained on the LAION-5B dataset. Our code is publicly available at `https://github.com/Subhodip123/VDU`.

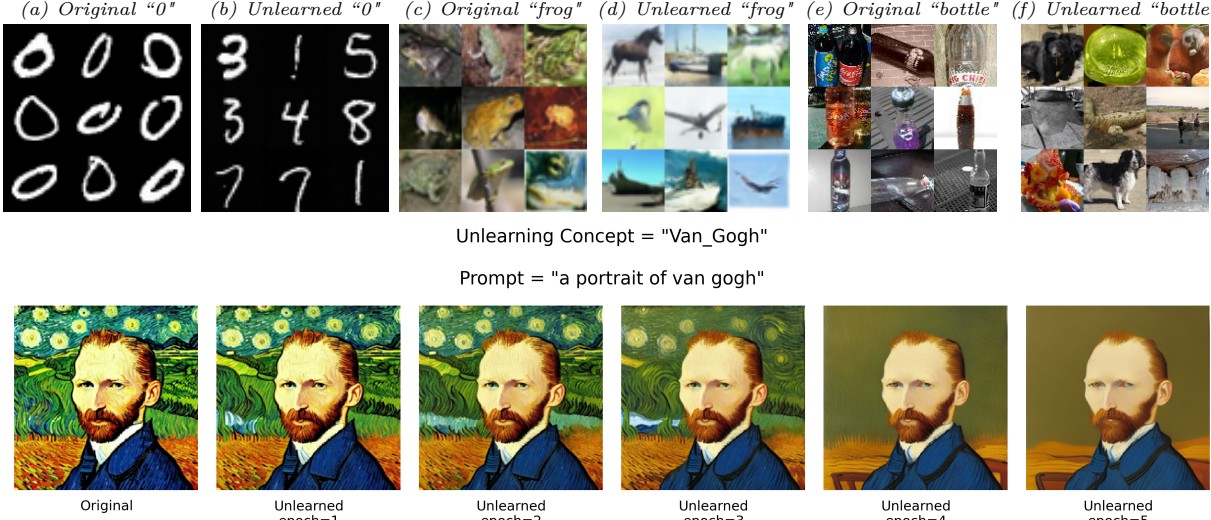

Figure 1: (a), (c), and (e) show the original images generated by a pre-trained DDPM model on the MNIST, CIFAR-10, and tinyImageNet datasets, respectively. (b), (d) and (f) display the corresponding images generated after unlearning, using our method **VDU**. The same noise vectors used to generate the original images were applied in the unlearned model to generate the unlearned images. The latter images show the performance of **VDU** for unlearning 'Van Gogh' style from a Stable Diffusion model. It is shown that the model slowly unlearns this artistic style feature over multiple epochs. **VDU** delivers good-quality images after unlearning, as well. Additional results on class-unlearning and feature-unlearning can be found in Figure 3, Figure 4 Figure 7, Figure 8, Figure 9, and Figure 10.

# 1 Introduction

In recent years, diffusion models (Sohl-Dickstein et al., 2015; Song & Ermon, 2019; Ho et al., 2020; Song et al., 2021; Rombach et al., 2022) have been popular for generating high-quality images that are useful for various tasks such as image and video editing (Ceylan et al., 2023; Feng et al., 2024), text-to-image translation (Ramesh et al., 2021; 2022; Saharia et al., 2022) etc. As these models become widespread, there lies a requirement to train them on vast amounts of internet data for diverse and robust output generation. However, there is also a potential downside to using such models, as they often generate outputs containing biased, violent, and obscene features (Tommasi et al., 2017). Thus, a safe and responsible generation from these models becomes an important requirement.

To address these challenges, recent works (Schramowski et al., 2023; Moon et al., 2024; Tiwary et al., 2023; Panda & A.P., 2023; Gandikota et al., 2023a; Heng & Soh, 2023; Fan et al., 2024) have proposed methods for regulating the outputs of various generative models (e.g. VAEs (Kingma, 2013), GANs (Goodfellow et al., 2020), and Diffusion models (Ho et al., 2020)) to ensure their safe and responsible deployment, with *machine unlearning* emerging as a particularly important technique for controlling the safe generation of content from these models. The key idea of *machine unlearning* is to develop a computationally efficient method to forget the subset of training data containing these undesired features from the pre-trained model. Thus, ideally, the unlearned model should behave like the retrained model — trained without the undesired data subset. However, achieving this goal is challenging because, during the process of unlearning, the model's generalization capacity gets hurt, making the quality of the generated outputs poor. This phenomenon is well studied in a similar context, also known as *catastrophic forgetting* (McCloskey & Cohen., 1989; Goodfellow et al., 2014; Kirkpatrick et al., 2017; Ginart et al., 2019; Nguyen et al., 2020) where plasticity to adapt to a new task hurts the stability of the model to perform well on the older task it had been trained on. In this case, the new task of unlearning an undesired subset of data hurts the quality of images generated by the unlearned model thereafter. Thus, the primary question of unlearning is:

*Can we develop an algorithm that can forget an undesired subset of training data from a pre-trained diffusion model while minimizing the degradation in generation quality?*

To answer this question, a recent unlearning work Selective Amnesia (SA) (Heng & Soh, 2023) adopts a continual learning setup and proposes an unlearning method based on elastic weight consolidation (EWC) (Kirkpatrick et al., 2017) wherein a weight regularization strategy, in the parameter space, is introduced to balance the unlearning task and to retain the sample quality. In essence, the variation of parameters is penalized by computing the "importance", to retain good performance, using the Fisher Information Matrix (FIM) (Sagun et al., 2017). However, this unlearning method has two potential downsides: (a) EWC formulation requires the calculation of the FIM, which is expensive due to the gradient product. (b) It often generates low-quality samples when it relies solely on the unlearning data subset. To solve the problem of low generation quality, the authors employ a generative replay to retrain the model with generated samples from the "non-unlearning" data subsets, further enhancing computational requirements. Similarly, Saliency Unlearning (SalUn) (Fan et al., 2024) proposes the perturbation of the most influential parameters across the entire model, with the assumption that the complete training dataset is available for reference. A major challenge, however, lies in situations with partial access to the training data due to rising concerns for data privacy and safety (Bae et al., 2018). In such a case, these methods performs poorly with no access to the non-unlearning data.

Taking note of such crucial observations, our research aims to develop a computationally efficient algorithm for unlearning an undesired class of training data from a pre-trained diffusion model. It is also important to mention that our methodology only requires partial access to a subset of training data aimed at unlearning because it is not always feasible to have access to the full training dataset (Chundawat et al., 2023b; Panda et al., 2024). While such a realistic setup is challenging, we draw inspiration from works on variational inference techniques (Nguyen et al., 2018; Noel Loo, 2021; Knoblauch et al., 2022; Wild et al., 2022). We propose, $\underline{V}$ariational $\underline{D}$iffusion $\underline{U}$nlearning (**VDU**) algorithm using a variational inference framework in the parameter space, to unlearn a subset of training data. The theoretical formulation of the variational divergence yields a lower bound that is adapted as a loss function to fine-tune the pre-trained model for unlearning. The proposed loss consists of two terms: *plasticity inducer* and *stability regularizer*. The *plasticity inducer* is used for adapting to the new task of reducing the log-likelihood of the unlearning data, while the *stability regularizer* prevents drastic changes in the pre-trained parameters of the model. These two proposed terms capture the persistent trade-offs that exist between the quantity of unlearning required and maintaining initial generation quality. Overall, our contributions are summarized as follows:

- One of our key contributions is proposing a variational inference framework for unlearning in diffusion models. Specifically, we formulate the unlearning objective as a variational divergence minimization and achieve class and feature unlearning in diffusion models.

- To address the limitations of concurrent unlearning methods (Heng & Soh, 2023; Fan et al., 2024) for diffusion models, which stem from assuming full access to the training dataset, our proposed method is more data-efficient. It is effective with fewer samples, requiring access only to the unlearning data points, and making it highly suitable for stricter unlearning scenarios with limited access to the original training dataset.

- We propose a methodology to unlearn different classes from a pre-trained unconditional Denoising Diffusion Probabilistic Model (DDPM) (Ho et al., 2020) trained on MNIST (LeCun et al., 1998), CIFAR-10 (Krizhevsky et al., 2009), and tinyImageNet (Le & Yang, 2015) datasets. We further validate our method for unlearning the generation of particular user-identified features from a pre-trained Stable Diffusion Model (Rombach et al., 2022) trained on the LAION-5B dataset (Schuhmann et al., 2022).

## 2 Related Works

### 2.1 Machine Unlearning for Generative Models

The core of machine unlearning (Cao & Yang, 2015; Xu et al., 2020; Nguyen et al., 2022; Bourtoule et al., 2021) revolves around removing or forgetting a specific subset of training data from a trained model. A

plausible approach is to retrain the model on the training data devoid of the undesired training data subset. However, this can be computationally very expensive concerning the scale of the model parameters and training data. To solve this problem, different machine unlearning algorithms were proposed for different problems and model settings, such as for K-means (Ginart et al., 2019), random forests (Brophy & Lowd, 2021), linear classification models (Guo et al., 2019; Golatkar et al., 2020a;b; Sekhari et al., 2021), neural network-based classifiers (Wu et al., 2020; Graves et al., 2021; Chundawat et al., 2023a; Panda et al., 2024). Further, machine unlearning is also useful for generative settings. With the emergence of large pre-trained text-to-image models (Rombach et al., 2022; Saharia et al., 2022), there is potential for misuse in generating harmful or inappropriate content. Thus, controlling the outputs from these generative models becomes an utmost priority. To solve this problem, recent works (Sun et al., 2023; Tiwary et al., 2023; Moon et al., 2024) proposed unlearning-based approaches for variational auto-encoders (VAEs) and generative adversarial networks (GANs). Sun et al. (2023) proposed a cascaded unlearning method using the idea of latent space substitution for a pre-trained StyleGAN under both settings of full and partial access (similar to our setting) to the training dataset. To extend unlearning for diffusion models, a recent work (Heng & Soh, 2023) adopts a continual learning setup and proposes an unlearning method based on elastic weight consolidation (EWC) (Kirkpatrick et al., 2017) for unlearning a pre-trained conditional DDPM (Ho et al., 2020). Further to enhance the performance of machine unlearning methods Fan et al. (2024) propose a 'weight saliency' method inspired by input saliency. This method focuses on certain influential weights to unlearn both in classification and generative models.

## 2.2 Variational Inference

To acquire exact inference from data, it is essential to calculate the exact posterior distribution. However, the exact posterior is often intractable and hard to calculate essentially making the inference task challenging. To solve this problem, the domain of variational inference tries to approximate the true posterior by a more tractable distribution from a class of distributions. Now to get the optimal distribution, often termed as variational posterior, these methods (Sato, 2001; Broderick et al., 2013; Blundell et al., 2015; Bui et al., 2016) optimize the so-called evidence lower bound (ELBO). These methodologies formulate the problem of inference in the parameter or weight space, which is often challenging because of the high dimension of the parameter space and multi-modality of parameter posterior distribution. Thus, to solve this problem, the recent line of works (Ma et al., 2019; Sun et al., 2019; Rudner et al., 2020; Wild et al., 2022) try to do inference in the function space itself. These methods (Rudner et al., 2020; Wild et al., 2022) perform inference by optimizing functional KL-divergence, minimizing the Wasserstein distance between the functional prior and the Gaussian process. Inspired by these works, we formulate our unlearning methodology from a task of inference in parameter space by minimizing a variational divergence.

# 3 Preliminaries

## 3.1 Unlearning Setup

Consider a pre-trained DDPM model, denoted as $f_{\theta^*}$, with initial parameters $\theta^* \in \Theta \subseteq \mathbb{R}^d$. $\Theta$ denotes the complete parameter space. This model has been trained on a specific training dataset $D$, consisting of $m$ i.i.d. samples $\{x_i\}_{i=1}^m$ that are drawn from a distribution $P_\mathcal{X}$ over the data space $\mathcal{X}$, tries to learn the underlying data distribution $P_\mathcal{X}$. Based on the outputs of this model, the user wants to unlearn a portion of the data space consisting of undesired features, referred to as $\mathcal{X}_f$. Therefore, the entire data space can be expressed as the union of $\mathcal{X}_r$ and $\mathcal{X}_f$, where $\mathcal{X} = \mathcal{X}_r \bigcup \mathcal{X}_f$ or $\mathcal{X}_r = \mathcal{X} \backslash \mathcal{X}_f$. We denote the distributions over $\mathcal{X}_f$ and $\mathcal{X}_r$ as $P_{\mathcal{X}_f}$ and $P_{\mathcal{X}_r}$, respectively. The objective of the unlearning mechanism is to output a sanitized model $\theta^u$ that does not produce outputs within the domain $\mathcal{X}_f$. This implies that the model should be trained to generate data samples conforming to the distribution $P_{\mathcal{X}_r}$ only. Assuming access to the whole training dataset, a computationally expensive approach to achieving this is by retraining the entire model from scratch using a dataset $D_r = \{x_i\}_{i=1}^s \sim P_{\mathcal{X}_r}^s$ or equivalently, $D_r = D \setminus D_f$, where $D_f = \{x_i\}_{i=1}^t \sim P_{\mathcal{X}_f}^t$. It is important to note that we do not have access to $D_r$ in our setting. Hence, the method of retraining becomes infeasible. Figure 2 illustrates our proposed approach.

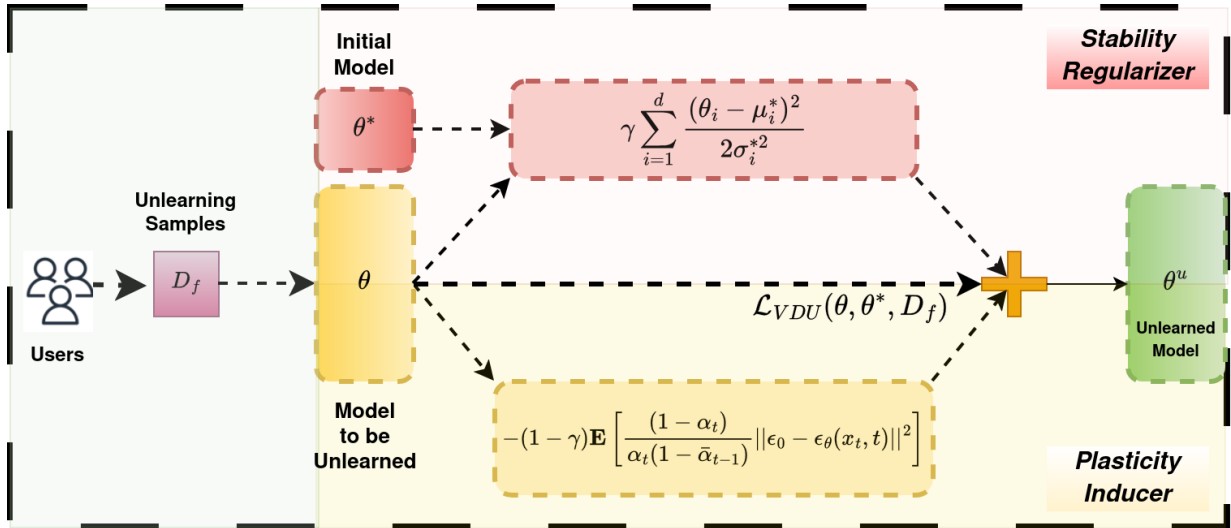

Figure 2: **Variational Diffusion Unlearning (VDU):** Given user-identified samples to be unlearned ($D_f$), the proposed VDU finetunes the initial model based on a two-term loss function. The first term, the ***Plasticity Inducer*** (shown in the bottom half), minimizes the log-likelihood for the unlearned samples to eliminate their influence, while the second term, the ***Stability Regularizer*** (shown in the upper half), preserves the model's overall performance on the remaining data.

## 3.2 Background on Diffusion Models

In diffusion models, data generation is formulated as a two-stage stochastic process. First, a forward diffusion process progressively corrupts clean data by adding noise according to a Markov kernel $q(.)$, resulting in a sequence of latent variables that increasingly resemble pure noise. This forward process is fixed and fully specified by a predefined noise schedule. Second, a reverse (denoising) process reconstructs data by iteratively removing noise using a learnable Markov kernel $p(.)$, which is typically parameterized by a neural network. Learning in diffusion models, therefore, amounts to approximating the reverse process so as to invert the fixed forward diffusion dynamics. Let $\{x_t : t = 0, 1, \ldots, T\}$ denote latent variables with $x_0$ denoting the true data. It is important to mention that in the diffusion process, it is assumed that all transitional kernels are first-order Markov. Now, in the forward diffusion process, the transition kernel is denoted as $q(x_t|x_{t-1})$ with the joint posterior distribution being $q(x_{1:T}|x_0) = \prod_{t=1}^{T} q(x_t|x_{t-1})$ where each $q(x_t|x_{t-1}) = \mathcal{N}(x_t; \sqrt{\alpha_t}x_{t-1}, (1-\alpha_t)I)$. Here, $\{\alpha_t : t \in T\}$ refers to the diffusion model's noise scheduler where $\bar{\alpha}_t = \prod_{j=1}^{t} \alpha_j$. Similarly, for the backward diffusion process, the transitional kernel is denoted as $p(x_{t-1}|x_t)$ with joint distribution $p(x_{0:T}) = p(x_T) \prod_{t=1}^{T} p_\theta(x_{t-1}|x_t)$ where, $p(x_T) = \mathcal{N}(x_T; 0, I)$. Thus, after optimizing the diffusion model, the sampling procedure is done by sampling Gaussian noise from $p(x_T)$ and iteratively running the denoising transitions $p_\theta(x_{t-1}|x_t)$ for $T$ steps to generate a new sample $x_0$. $\epsilon_0$ is the true added noise, $\epsilon_\theta(x_t, t)$ is the model predicted noise at time $t$.

## 4 Unlearning as Variational Divergence Minimization

Given the pre-trained DDPM model $f_{\theta^*}$ and unlearning data subset $D_f$, the objective is to produce an unlearned model $\theta^u$ so that it behaves like a retrained model $\theta^r$ trained on $D_r$. Inspired by some previous works in Bayesian inference (Sato, 2001; Broderick et al., 2013; Blundell et al., 2015; Ghahramani & Attias, 2020; Nguyen et al., 2020), it can be seen that, retrained parameters $\theta^r$ are a sample from the retrained model's parameter posterior distribution $P(\theta|D_r)$ i.e., $\theta^r \sim P(\theta|D_r)$. Similarly, the pre-trained model's parameters $\theta^* \sim P(\theta|D_r, D_f)$. Using this motivation, we try to approximate the retrained model's parameter posterior distribution $P(\theta|D_r)$ as follows: $P(\theta|D_r) \propto \frac{P(\theta|D_r, D_f)}{P(D_f|\theta)}$. This is true by applying Bayes' rule (ignoring the normalizing constant) and a simple consequence of i.i.d assumption of data in $D_r$ and $D_f$.

It can be seen that the posterior distribution $P(\theta|D_r)$ is intractable, and an approximation is required by forming $proj(P(\theta|D_r)) \approx Q^*(\theta)$. Here, $proj(\cdot)$ is a projection function that takes an intractable unnormalized distribution and maps it to a normalized distribution. As previously mentioned in the literature (Broderick et al., 2013; Blundell et al., 2015; Bui et al., 2016; Nguyen et al., 2018; Noel Loo, 2021; Knoblauch et al., 2022; Wild et al., 2022), one can take several choices of projection functions, such as Laplace's approximation, variational KL-divergence minimisation, moment matching, and importance sampling. Here, we consider the variational KL-divergence minimization for our method because previous works (Bui et al., 2016) noted that it outperforms other inference techniques for complex models. Thus, our method is defined through a variational KL-divergence minimization over a set of probable approximate posterior distributions $\mathcal{Q}$ as follows:

$$Q^*(\theta) = \underset{Q(\theta) \in \mathcal{Q}}{\text{argmin}} D_{KL} \left( Q(\theta) \middle\| Z \cdot \frac{P(\theta|D_f, D_r)}{P(D_f|\theta)} \right) \tag{1}$$

Here $Z$ is the intractable normalization constant, which is independent of the parameter $\theta$. Finally, taking the Gaussian mean-field assumption in the parameter space i.e. if the variational posterior distribution $Q(\theta) = \prod_{i=1}^{d} \mathcal{N}(\theta_i, \sigma_i^2)$ and the posterior distribution with full data $P(\theta|D_r, D_f) = \prod_{i=1}^{d} \mathcal{N}(\mu_i^*, \sigma_i^{*2})$, Eq. 1 reduces to minimizing the following loss function for a diffusion model. We refer this loss as the Variational Diffusion Unlearning (VDU) loss, which is defined as follows.

$$\mathcal{L}_{VDU}(\theta, \theta^*, D_f) = \underbrace{-(1-\gamma) \underset{\substack{x_0 \sim D_f \\ \epsilon_0, t}}{\mathbb{E}} \left[ \frac{(1-\alpha_t)}{\alpha_t(1-\bar{\alpha}_{t-1})} ||\epsilon_0 - \epsilon_\theta(\sqrt{\bar{\alpha}_t}x_0 + \sqrt{1-\bar{\alpha}_t}\epsilon_0, t)||^2 \right]}_{A} + \underbrace{\gamma \sum_{i=1}^{d} \frac{(\theta_i - \mu_i^*)^2}{2\sigma_i^{*2}}}_{B} \tag{2}$$

Here $\gamma$ is a hyper-parameter, $d$ is the model dimension. Eq. 2 represents the proposed loss function used to optimize the pre-trained model during unlearning. The loss comprises two key terms: $A$ is referred to as the **plasticity inducer**, and minimizes the log-likelihood of the unlearning data. In contrast, the term $B$, serving as the **stability regularizer**, penalizes the model's parameters to prevent them from deviating too much from their pre-trained state during unlearning. To balance these two components, we introduce a hyper-parameter $\gamma$. Figure 2 is a detailed illustration of our framework and details the different

> **Variational Diffusion Unlearning**
>
> **Input:** $D_f$, $\theta^*$, $E$, $\eta$, $\gamma$
> **Initialize:** $\theta \leftarrow \theta^*$
> 1: **for** $t = 1$ to $E$ **do**
> 2:     Compute $\mathcal{L}_{VDU}(\theta, \theta^*, D_f)$
> 3:     Update $\theta \leftarrow \theta - \eta \nabla_\theta \mathcal{L}_{VDU}$
> 4: **end for**
> **Output:** $\theta^E$

loss components used for the unlearning of $D_f$. Further, the estimation of $\mu_i^*$ and $\sigma_i^*$ is discussed in Appendix A.3.2.

## 4.1 Theoretical Explanation for the Unlearning Loss

In this section, we provide a theoretical exposition for the derivation of the variational diffusion unlearning loss $\mathcal{L}_{VDU}(\theta, \theta^*, D_f)$ in Eq. 2 from variational divergence minimization defined in Eq. 1. This derivation is essentially presented as a key theorem in Theorem 1. The proof of this theorem essentially relies on the variational perspective of diffusion models, which is stated using Lemma 1 and Lemma 2.

**Lemma 1** *The log-likelihood under the backward diffusion process kernel,*

$$\ln p_\theta(x_0) \gtrsim -\sum_{t=2}^{T} \underset{q(x_t|x_0)}{\mathbb{E}} \left[ D_{KL}(q(x_{t-1}|x_t, x_0)||p_\theta(x_{t-1}|x_t)) \right] \tag{3}$$

**Lemma 2** *Assuming all the transition kernels to be Gaussian, the following hold:*

$$q(x_{t-1}|x_t, x_0) = \mathcal{N}(x_{t-1}; \mu_q(t), \sigma_q^2(t)I) \text{ with } \mu_q(t) = \frac{1}{\sqrt{\alpha_t}}x_t - \frac{1-\alpha_t}{\sqrt{1-\bar{\alpha}_t}\sqrt{\alpha_t}}\epsilon_0 \tag{4}$$

$$p_\theta(x_{t-1}|x_t) = \mathcal{N}(x_{t-1}; \mu_\theta(t), \sigma_q^2(t)I) \text{ with } \mu_\theta(t) = \frac{1}{\sqrt{\alpha_t}}x_t - \frac{1-\alpha_t}{\sqrt{1-\bar{\alpha}_t}\sqrt{\alpha_t}}\epsilon_\theta(x_t, t) \tag{5}$$

$$\sigma_q^2(t) = \frac{(1-\alpha_t)(1-\bar{\alpha}_{t-1})}{(1-\bar{\alpha}_t)} \tag{6}$$

Proof for the above two lemmas (with different notations) can be found in the existing literature (Sohl-Dickstein et al., 2015; Ho et al., 2020; Luo, 2022). For completeness, we provide the proofs in Appendix A.1.1 and A.1.2.

**Theorem 1** *Assuming a Gaussian mean-field approximation in the parameter space i.e., if the variational posterior distribution $Q(\theta) = \prod_{i=1}^d \mathcal{N}(\theta_i, \sigma_i^2)$ and the posterior distribution with full data $P(\theta|D_r, D_f) = \prod_{i=1}^d \mathcal{N}(\mu_i^*, \sigma_i^{*2})$, then*

$$D_{KL}\left(Q(\theta)\middle\|Z \cdot \frac{P(\theta|D_f, D_r)}{P(D_f|\theta)}\right) \gtrsim \underbrace{-\sum_{x_0 \in D_f}\sum_{t=2}^T \mathop{\mathbb{E}}_{q(x_t|x_0)}\left[\frac{(1-\alpha_t)}{\alpha_t(1-\bar{\alpha}_{t-1})}\|\epsilon_0 - \epsilon_\theta(x_t, t)\|^2\right]}_{I}$$

$$+ \underbrace{\sum_{i=1}^d\left[\frac{(\theta_i - \mu_i^*)^2}{2\sigma_i^{*2}} + \frac{\sigma_i^2}{2\sigma_i^{*2}} + \log\frac{\sigma_i^*}{\sigma_i} - \frac{1}{2}\right]}_{II} \tag{7}$$

**Proof 1** *Here, we give a rough sketch of the proof. The KL-divergence term in Eq. 1 is expanded and segregated into two terms:*

$$D_{KL}\left(Q(\theta)\middle\|Z \cdot \frac{P(\theta|D_f, D_r)}{P(D_f|\theta)}\right) = \mathop{\mathbb{E}}_{Q(\theta)}\left[\ln\frac{Q(\theta)P(D_f|\theta)}{Z \cdot P(\theta|D_f, D_r)}\right] \tag{8}$$

$$\stackrel{(g)}{=} \mathop{\mathbb{E}}_{Q(\theta)}\left[\ln P(D_f|\theta)\right] + \mathop{\mathbb{E}}_{Q(\theta)}\left[\ln\frac{Q(\theta)}{P(\theta|D_f, D_r)}\right] \tag{9}$$

$$\stackrel{(h)}{=} \mathop{\mathbb{E}}_{Q(\theta)}\left[\sum_{x_0 \in D_f}\ln P(x_0|\theta)\right] + \mathop{\mathbb{E}}_{Q(\theta)}\left[\ln\frac{Q(\theta)}{P(\theta|D_f, D_r)}\right] \tag{10}$$

$$= \underbrace{\mathop{\mathbb{E}}_{\theta \sim Q(\theta)}\left[\sum_{x_0 \in D_f}\ln P(x_0|\theta)\right]}_{\text{ⓐ}} + \underbrace{D_{KL}(Q(\theta)\|P(\theta|D_f, D_r))}_{\text{ⓑ}} \tag{11}$$

*(g) holds as the normalization constant is independent of $\theta$. (h) is true because of the i.i.d. assumption on the data. We further derive the term ⓐ $:= \mathbb{E}[\sum_{x_0 \in D_f} \ln P(x_0|\theta)]$ using Lemmas 1 and 2. While the second term ⓑ $:= D_{KL}(Q(\theta)\|P(\theta|D_r, D_f))$ is expanded using the assumption of KL divergence between two Gaussian distributions. For a detailed proof, please look into Appendix I.A.1.3.*

**Remark 1** *From Theorem 1, the first term $I$ is further adapted and used in the first part of the loss function $\mathcal{L}_{VDU}(\theta, \theta^*, D_f)$ in Eq. 2. Similarly, the second term $II$ is further simplified with additional assumptions to obtain the second part of the loss. These details are included in Appendix A.1.4*

## 5 Experiments and Results

### 5.1 Datasets and Models

The primary goal of our proposed unlearning method is to unlearn both class-level information and feature-level information. For class unlearning, our goal is to restrict the generation of samples from a certain

class. For this, we utilize unconditional DDPM (Ho et al., 2020) pre-trained on three well-known datasets: MNIST (LeCun et al., 1998), CIFAR-10 (Krizhevsky et al., 2009), and tinyImageNet (Le & Yang, 2015). Now for feature unlearning, we unlearn the generation of certain high-level features from a Stable Diffusion model (Rombach et al., 2022) pre-trained on the LAION-5B dataset (Schuhmann et al., 2022). More details about the model architectures and datasets can befound in Appendix A.4.1.

## 5.2 Pre-Training, Unlearning and Baselines

- **Pre-Training:** For class-unlearning setting, in order to achieve the pre-trained model, we train the unconditional DDPM model on MNIST, CIFAR-10, and tinyImageNet (tIN) datasets. These pre-trained models achieve FID (Heusel et al., 2017) scores of 5.12 on MNIST, 7.96 on CIFAR-10, and 18.92 on tinyImageNet datasets. For the feature-unlearning setting, we use pre-trained checkpoints of Stable Diffusion model from an open-sourced repository. A detailed description of the pre-training setup can be found in Appendix A.4.2.

- **Unlearning Setup:** Due to the unavailability of the total training dataset, our experimental setting is restricted to partial access, i.e., only to the data points belonging to the unlearning class. For class-unlearning, we unlearn multiple classes from MNIST, CIFAR-10, and tinyImageNet datasets. Now, for feature-unlearning, we try to unlearn some high-level features, such as an artistic style from a state-of-the-art Stable Diffusion model. Further experimental details of our unlearning method on each dataset and model are added in Appendix A.4.3.

- **Baselines:** If we assume that we have access to $D_r$ then retraining is the most optimal method. Due to its high computational cost, we fine-tune the model assuming access to $D_r$. Each initial model is fine-tuned with $D_r$ for a single epoch since it serves as an optimal baseline when comparing various approaches. Further, we compare our proposed method against several unlearning baselines tested for diffusion models: Selective Amnesia (SA) (Heng & Soh, 2023), Saliency Unlearning (SalUn) (Fan et al., 2024), and Erasing Stable Diffusion (ESD) (Gandikota et al., 2023a). Note that for a fair comparison, the loss component regarding $D_r$ is removed from all the above methods as we don't have access to $D_r$. Additionally, to account for the low computational cost of unlearning methods, the unlearning step for each method is only executed for very few epochs ($10\times$ of fine-tuning time), and the best outcomes among these few epochs are reported for each method. The reason for putting such strict constraints is discussed in detail in Appendix A.3.4.

## 5.3 Evaluation Metrics

- **Percentage of Unlearning (PUL):** Proposed by Tiwary et al. (2023), this metric measures how much unlearning has occurred by comparing the reduction in the number of unwanted samples produced by the model after unlearning ($\theta^u$) with the number of such samples before unlearning ($\theta^*$). The Percentage of Unlearning (PUL) is calculated as: $\text{PUL} = \frac{(D_f^g)_{\theta^*} - (D_f^g)_{\theta^u}}{(D_f^g)_{\theta^*}} \times 100\%$ where $(D_f^g)_{\theta^*}$ and $(D_f^g)_{\theta^u}$ represent the number of undesired samples generated by the original and the unlearned model, respectively. To calculate PUL, we generate 5,000 random samples from the unconditional DDPM model and use a pre-trained classifier to identify the unwanted samples. Details of this classifier are given in the Appendix A.4.4.

- **Unlearned Fréchet Inception Distance (u-FID):** To quantify the quality of generated images by the unlearned model, we utilize the u-FID score. It is important to mention that this FID score is measured between the generated samples from the unlearned model and $D_r$. Thus, in the case of unconditional DDPM, to remove the unlearning data points from the non-unlearning data, we again use the same pre-trained classifier. In this case, a lower u-FID score reflecting higher image quality indicates that the unlearned model's performance does not degrade on the non-unlearning data points.

## 5.4 Experimental Results

### 5.4.1 Quantitative and Qualitative Results

In Table 1, we contrast our method, VDU, against the above-mentioned baseline methods for different class unlearning settings. Overall, for all datasets and unlearned classes, it is observable that our method achieves lower u-FID scores with comparable to or superior PUL scores. Specifically, on the MNIST dataset, VDU

Table 1: PUL($\uparrow$) and u-FID($\downarrow$) scores (mean over three random seeds) comparisons between VDU and different unlearning baselines for class unlearning on the MNIST, CIFAR-10, and tinyImageNet (tIN) datasets.

| Datasets | Unlearned Classes | Fine-tuning | | SA | | SalUn | | ESD | | VDU (Ours) | |
|---|---|---|---|---|---|---|---|---|---|---|---|
| | | PUL (%) | u-FID | PUL (%) | u-FID | PUL (%) | u-FID | PUL (%) | u-FID | PUL (%) | u-FID |
| **MNIST** | Digit-1 | 90.07 | 5.90 | 15.01 | 301.26 | 45.28 | 68.75 | 56.66 | 66.15 | **75.06** | **14.43** |
| | Digit-8 | 94.29 | 6.15 | 48.95 | 161.09 | 32.53 | 50.18 | 32.68 | 55.67 | **68.96** | **38.20** |
| **CIFAR-10** | Automobile | 88.01 | 9.02 | 2.25 | 92.89 | 12.01 | 66.47 | 8.97 | 73.48 | **60.87** | **30.86** |
| | Ship | 92.86 | 10.74 | **85.02** | 249.89 | 55.13 | 75.48 | 58.27 | 77.12 | 71.63 | **24.46** |
| **tIN** | Fish | 82.56 | 13.75 | **67.7** | 32.00 | 39.33 | 42.70 | 41.67 | 45.19 | 58.75 | **29.92** |
| | Butcher Shop | 83.37 | 16.82 | 6.7 | 24.00 | 2.87 | 33.47 | 8.76 | 39.48 | **66.67** | **24.17** |

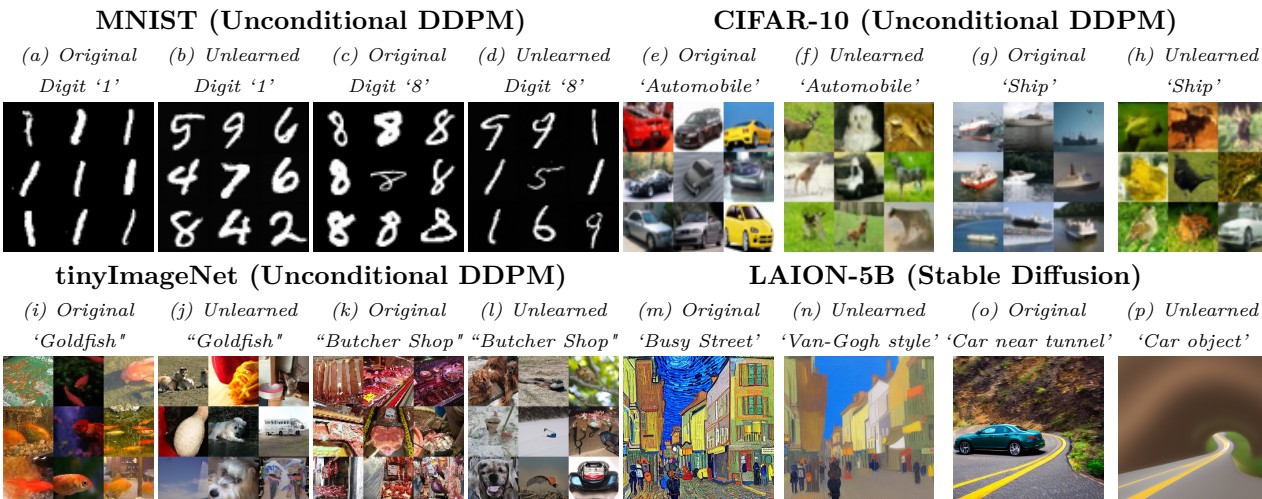

**MNIST (Unconditional DDPM)**       **CIFAR-10 (Unconditional DDPM)**

*(a) Original Digit '1'*  *(b) Unlearned Digit '1'*  *(c) Original Digit '8'*  *(d) Unlearned Digit '8'*  *(e) Original 'Automobile'*  *(f) Unlearned 'Automobile'*  *(g) Original 'Ship'*  *(h) Unlearned 'Ship'*

**tinyImageNet (Unconditional DDPM)**       **LAION-5B (Stable Diffusion)**

*(i) Original 'Goldfish"*  *(j) Unlearned "Goldfish"*  *(k) Original "Butcher Shop"*  *(l) Unlearned "Butcher Shop"*  *(m) Original 'Busy Street'*  *(n) Unlearned 'Van-Gogh style'*  *(o) Original 'Car near tunnel'*  *(p) Unlearned 'Car object'*

Figure 3: Generated samples from the pre-trained model and different unlearned models. (b),(d),(f),(h),(j) & (l) are generated using the same noise vectors from unlearned unconditional DDPM on MNIST, CIFAR-10, and tinyImageNet datasets. (m)–(p) show Stable Diffusion samples before and after unlearning the 'Van Gogh style' and 'Car object' using prompts from LAION-5B. More results are provided in Figure 7 and Figure 8 in Appendix.

outperforms all other baseline methods, with large improvements in u-FID, after just 1 epoch of training. Similarly, for tinyImageNet, VDU achieves high PUL scores for unlearning "Butcher Shop" compared to all other baseline methods. To summarize, it can be seen that our method retains superior to comparable PUL with superior u-FID scores across a variety of class unlearning scenarios over multiple datasets. Similarly, the performance of VDU for unlearning features from a pre-trained Stable Diffusion model is shown in Figure 3. This shows that VDU is applicable not only for low-dimensional settings but also works well in high-dimensional datasets trained on state-of-the-art Stable Diffusion models. More experimental results can be found in Appendix A.2.

### 5.4.2 Ablation Results

**Effect of Unlearning Epochs:** Table 2 reports the performance of our method across different unlearning epochs. We observe that increasing the number of unlearning epochs improves the degree of unlearning, while simultaneously degrading the generation quality. This behavior can be intuitively explained by our objective function: the first term of the loss is unbounded, and continued minimization increasingly suppresses the

Table 2: Effect of training epochs on unlearning performance measured on VDU

| Dataset | Unlearned Class | Epoch 1 | | Epoch 2 | | Epoch 3 | | Epoch 4 | | Epoch 5 | |
|---|---|---|---|---|---|---|---|---|---|---|---|
| | | PUL (%) | u-FID | PUL (%) | u-FID | PUL (%) | u-FID | PUL (%) | u-FID | PUL (%) | u-FID |
| MNIST | Digit-1 | 75.06 | 14.43 | 86.58 | 33.67 | 97.86 | 56.17 | 99.55 | 106.18 | 100.00 | 218.83 |
| CIFAR-10 | Automobile | 27.33 | 26.82 | 43.67 | 30.19 | 50.58 | 29.82 | 60.87 | 30.86 | 68.97 | 56.82 |
| tinyImageNet | Fish | 33.26 | 18.72 | 58.75 | 29.92 | 59.86 | 37.68 | 59.88 | 52.72 | 60.65 | 70.95 |

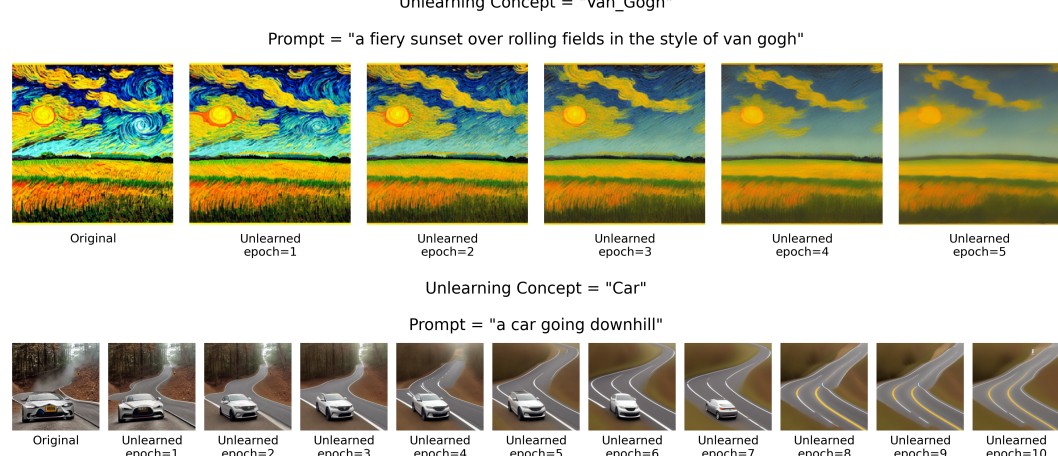

Figure 4: Results of VDU on unlearning *'Van Gogh style'* and *'Car object'* over multiple epochs

targeted features, thereby enhancing unlearning. However, this aggressive optimization drives the model further away from the original model in parameter space, leading to a deterioration in generation quality. Figure 4 visually highlights this inherent trade-off between unlearning effectiveness and generative fidelity. For these experiments, we set the gamma to be 0.3 for all the datasets. We see a similar trend in the visual results of Figure 8, Figure 9, and Figure 10 for feature unlearning in Stable Diffusion.

**Contribution of $\gamma$:** We evaluate the reliance of our method on the hyper-parameter $\gamma \in [0, 1]$. In Table 3, we illustrate the impacts of varying $\gamma$ on MNIST (unlearning the digit "1"), CIFAR-10 (unlearning "Ship"), and tinyImageNet (unlearning "Butcher Shop"). It can be seen that, with increasing $\gamma$, Eq. 2 tends towards increasing the strength of the *stability regularizer*. This leads to the the model degenerating into the pre-

Table 3: Impact of $\gamma$ on the unlearning performance of VDU.

| $\gamma$ | MNIST (Digit-1) | | CIFAR-10 (Ship) | | tIN (Butcher Shop) | |
|---|---|---|---|---|---|---|
| | PUL | u-FID | PUL | u-FID | PUL | u-FID |
| 0 | 67.98 | 13.74 | 70.89 | 45.72 | 68.79 | 25.74 |
| 0.1 | 75.06 | 14.43 | 71.63 | 24.46 | 60.10 | 27.84 |
| 0.3 | 72.15 | 14.33 | 74.65 | 28.75 | 55.65 | 21.17 |
| 0.6 | 55.21 | 11.54 | 56.54 | 20.51 | 66.67 | 24.17 |
| 0.8 | 71.67 | 13.12 | 69.95 | 19.64 | 58.86 | 20.92 |
| 1 | 16.84 | 7.48 | 1.67 | 13.16 | 20.56 | 21.08 |

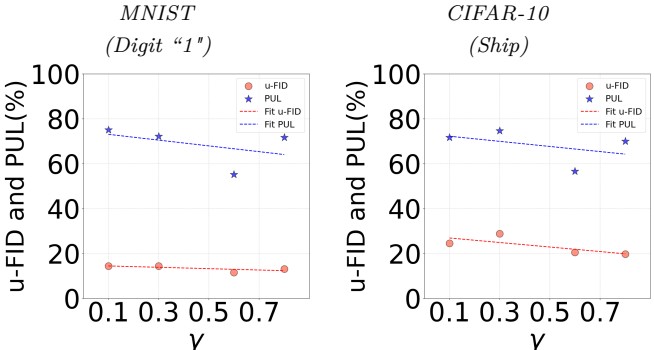

Figure 5: Ablation results on varying $\gamma$ for class unlearning of MNIST and CIFAR-10 datasets. Increasing value of $\gamma$ results in decrease in PUL and u-FID metrics.

trained model. Due to this, we observe a steady decline in PUL since the proposed loss attends to preserving the generated sample quality only. Such a trend is also bolstered by the lowering of the u-FID, indicating good sample quality. A visual representation of these trends is depicted in Figure 5.

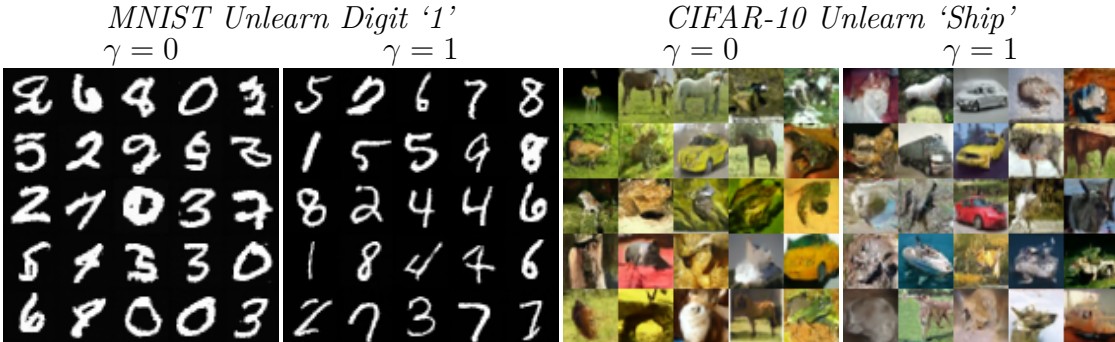

Figure 6: Visual illustrations of generated image samples with $\gamma = 0$ and $\gamma = 1$ on MNIST and CIFAR-10 for the class unlearning setting.

**Nullify Loss Components:** Nullifying loss components with $\gamma=0$ and $\gamma=1$ for unlearning Digit-1 in MNIST and Ship class in CIFAR-10 are shown visually in Figure 6. We observe that when $\gamma=0$ or when the *stability regularizer* is nullified, the quality of generated samples drastically declines and images of the Digit-1 and Ship classes are not generated. On the other hand, image quality increases when $\gamma=1$, but class-specific information is still preserved, leading to the generation of Digit-1 and Ship classes. Overall, by varying $\gamma$, our proposed loss strives to achieve a balance between unlearning and preserving image quality.

## 6 Conclusion, Limitations and Future works

In this paper, we present Variational Diffusion Unlearning (**VDU**), a machine unlearning framework to prevent the generation of undesired outputs from a pre-trained diffusion model, with sole access to the training data subset to be unlearned. Our current theoretical framework leverages the parameter space to exploit variational inference by minimizing a loss function comprising two key components: the *plasticity inducer* and the *stability regularizer*. While the *plasticity inducer* focuses on minimizing the log-likelihood of the undesired training samples, the *stability regularizer* preserves the quality of the generated image samples by regularizing the model in the parameter space. Through comprehensive experiments, we demonstrate our method's effectiveness in different class unlearning settings for the MNIST, CIFAR-10, and tinyImageNet datasets. Further, the effectiveness of our approach is bolstered by various feature unlearning from state-of-the-art Stable Diffusion models pre-trained on LAION-5B dataset. This highlights that under restricted access to the training dataset, our method is effective for both low and high-dimensional settings, making it a practical choice unlearning from diffusion models.

**Limitations and Future works:** Apart from its effectiveness, we highlight some key limitations, such as our current framework assumes a Gaussian mean-field approximation on the parameter distributions that might not always be valid. Thus, exploration of more generic distributional settings is left as a part of our future work. Also, additional empirical evaluation as per the recent unlearning benchmarking dataset (Zhang et al., 2024) is a key limitation and left as a part of future work. Further to overcome the limited theoretical scope of variational inference in parameter space, a broader theoretical framework using function space variational inference techniques (Ma et al., 2019; Rudner et al., 2020; Sun et al., 2019; Wild et al., 2022), will also be explored in our future work.

## Acknowledgments

Subhodip, a current Ph.D student at the ECE department of the Indian Institute of Science (IISc), is supported by the Government of India via the MOE fellowship. Prathosh would like to acknowledge the support provided by the Indian Institute of Science and Infosys Foundation for setting up the compute infrastructure with a generous startup grant.

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

# A  Technical Appendices and Supplementary Material

## A.1  Theoretical Analysis

### A.1.1  Proof of Lemma-1

Let $x_0$ denote the true data. Thus, to increase the log-likelihood of the data, we maximize the evidence lower bound as follows:

$$\ln p_\theta(x_0) = \ln \int p_\theta(x_{0:T}) \, dx_{1:T} \tag{12}$$

$$= \ln \int \frac{p_\theta(x_{0:T})}{q(x_{1:T}|x_0)} q(x_{1:T}|x_0) \, dx_{1:T} \tag{13}$$

$$= \ln \mathbb{E}_{q(x_{1:T}|x_0)} \left[ \frac{p_\theta(x_{0:T})}{q(x_{1:T}|x_0)} \right] \tag{14}$$

$$\overset{(a)}{\geq} \mathbb{E}_{q(x_{1:T}|x_0)} \left[ \ln \frac{p_\theta(x_{0:T})}{q(x_{1:T}|x_0)} \right] \tag{15}$$

$$= \mathbb{E}_{q(x_{1:T}|x_0)} \left[ \ln \frac{p(x_T) \prod_{t=1}^{T} p_\theta(x_{t-1}|x_t)}{\prod_{t=1}^{T} q(x_t|x_{t-1})} \right] \tag{16}$$

$$= \mathbb{E}_{q(x_{1:T}|x_0)} \left[ \ln \frac{p(x_T) p_\theta(x_0|x_1) \prod_{t=2}^{T} p_\theta(x_{t-1}|x_t)}{q(x_1|x_0) \prod_{t=2}^{T} q(x_t|x_{t-1})} \right] \tag{17}$$

$$\overset{(b)}{=} \mathbb{E}_{q(x_{1:T}|x_0)} \left[ \ln \frac{p(x_T) p_\theta(x_0|x_1) \prod_{t=2}^{T} p_\theta(x_{t-1}|x_t)}{q(x_1|x_0) \prod_{t=2}^{T} q(x_t|x_{t-1}, x_0)} \right] \tag{18}$$

$$= \mathbb{E}_{q(x_{1:T}|x_0)} \left[ \ln \frac{p_\theta(x_T) p_\theta(x_0|x_1)}{q(x_1|x_0)} + \ln \prod_{t=2}^{T} \frac{p_\theta(x_{t-1}|x_t)}{q(x_t|x_{t-1}, x_0)} \right] \tag{19}$$

$$\overset{(c)}{=} \mathbb{E}_{q(x_{1:T}|x_0)} \left[ \ln \frac{p(x_T) p_\theta(x_0|x_1)}{q(x_1|x_0)} + \ln \prod_{t=2}^{T} \frac{p_\theta(x_{t-1}|x_t)}{\frac{q(x_{t-1}|x_t, x_0) q(x_t|x_0)}{q(x_{t-1}|x_0)}} \right] \tag{20}$$

$$= \mathbb{E}_{q(x_{1:T}|x_0)} \left[ \ln \frac{p(x_T) p_\theta(x_0|x_1)}{q(x_1|x_0)} + \ln \frac{q(x_1|x_0)}{q(x_T|x_0)} + \ln \prod_{t=2}^{T} \frac{p_\theta(x_{t-1}|x_t)}{q(x_{t-1}|x_t, x_0)} \right] \tag{21}$$

$$= \mathbb{E}_{q(x_{1:T}|x_0)} \left[ \ln \frac{p(x_T) p_\theta(x_0|x_1)}{q(x_T|x_0)} + \sum_{t=2}^{T} \ln \frac{p_\theta(x_{t-1}|x_t)}{q(x_t - 1|x_t, x_0)} \right] \tag{22}$$

$$= \mathbb{E}_{q(x_{1:T}|x_0)} \left[ \ln p_\theta(x_0|x_1) \right] + \mathbb{E}_{q(x_{1:T}|x_0)} \left[ \ln \frac{p(x_T)}{q(x_T|x_0)} \right] + \sum_{t=2}^{T} \mathbb{E}_{q(x_{1:T}|x_0)} \left[ \ln \frac{p_\theta(x_{t-1}|x_t)}{q(x_{t-1}|x_t, x_0)} \right] \tag{23}$$

$$\overset{(d)}{=} \mathbb{E}_{q(x_1|x_0)} \left[ \ln p_\theta(x_0|x_1) \right] + \mathbb{E}_{q(x_T|x_0)} \left[ \ln \frac{p(x_T)}{q(x_T|x_0)} \right] + \sum_{t=2}^{T} \mathbb{E}_{q(x_t, x_{t-1}|x_0)} \left[ \ln \frac{p_\theta(x_{t-1}|x_t)}{q(x_{t-1}|x_t, x_0)} \right] \tag{24}$$

$$\overset{(e)}{=} \mathbb{E}_{q(x_1|x_0)} \left[ \ln p_\theta(x_0|x_1) \right] - D_{KL}(q(x_T|x_0)||p(x_T)) \tag{25}$$

$$- \sum_{t=2}^{T} \mathbb{E}_{q(x_t|x_0)} \left[ D_{KL}(q(x_{t-1}|x_t, x_0)||p_\theta(x_{t-1}|x_t)) \right] \tag{26}$$

$$\overset{(f)}{\approx} - \sum_{t=2}^{T} \mathbb{E}_{q(x_t|x_0)} \left[ D_{KL}(q(x_{t-1}|x_t, x_0)||p_\theta(x_{t-1}|x_t)) \right] \tag{27}$$

Here $(a)$ holds via Jensen's inequality as log is a concave function. $(b)$ is true because additional conditioning doesn't affect the first-order Markovian transitional kernel $q(.|.)$. $(c)$ and $(e)$ are achieved via Bayes' rule

while $(d)$ is true via marginalization property. As the two terms in Eq. 25 are insignificant compared to the term in Eq. 26, so $(f)$ holds.

### A.1.2 Proof of Lemma-2

Even though it is a three-part proof, part-(3) comes from the proof of part-(1) while part-(2) is proved via a similar argument as the proof of part-(1). So, here we will prove part-(1) in detail. We know:

$$q(x_t|x_{t-1}, x_0) = q(x_t|x_{t-1}) = (x_t; \sqrt{\alpha_t}x_{t-1}, (1-\alpha_t)I) \tag{28}$$

Now, we can represent $x_t$ in terms of $x_0$ by recursive re-parameterization as,

$$x_t = \sqrt{\alpha_t}x_{t-1} + \sqrt{1-\alpha_t}\epsilon_{t-1} \tag{29}$$

$$= \sqrt{\alpha_t}\left(\sqrt{\alpha_{t-1}}x_{t-2} + \sqrt{1-\alpha_{t-1}}\epsilon_{t-2}\right) + \sqrt{1-\alpha_t}\epsilon_{t-1} \tag{30}$$

$$= \sqrt{\alpha_t\alpha_{t-1}}x_{t-2} + \sqrt{\alpha_t - \alpha_t\alpha_{t-1}}\epsilon_{t-2} + \sqrt{1-\alpha_t}\epsilon_{t-1} \tag{31}$$

$$= \sqrt{\alpha_t\alpha_{t-1}}x_{t-2} + \sqrt{\left(\sqrt{\alpha_t - \alpha_t\alpha_{t-1}}\right)^2 + \sqrt{(1-\alpha_t)^2}}\epsilon_{t-2} \tag{32}$$

$$= \sqrt{\alpha_t\alpha_{t-1}}x_{t-2} + \sqrt{\alpha_t - \alpha_t\alpha_{t-1} + 1 - \alpha_t}\epsilon_{t-2} \tag{33}$$

$$= \sqrt{\alpha_t\alpha_{t-1}}x_{t-2} + \sqrt{1-\alpha_t\alpha_{t-1}}\epsilon_{t-2} \tag{34}$$

$$\vdots \tag{35}$$

$$= \sqrt{\prod_{i=1}^{t}\alpha_i}x_0 + \sqrt{1 - \prod_{i=1}^{t}\alpha_i}\epsilon_0 \tag{36}$$

$$= \sqrt{\bar{\alpha}_t}x_0 + \sqrt{1-\bar{\alpha}_t}\epsilon_0 \tag{37}$$

Thus, $x_t \sim \mathcal{N}(x_t; \sqrt{\bar{\alpha}_t}x_0, (1-\bar{\alpha}_t)I)$. Now, via Bayes' rule:

$$q(x_{t-1}|x_t, x_0) = \frac{q(x_t|x_{t-1}, x_0)q(x_{t-1}|x_0)}{q(x_t|x_0)} \tag{38}$$

$$= \frac{\mathcal{N}(x_t; \sqrt{\alpha_t}x_{t-1}, (1-\alpha_t)I)\,\mathcal{N}(x_{t-1}; \sqrt{\bar{\alpha}_{t-1}}x_0, (1-\bar{\alpha}_{t-1})I)}{\mathcal{N}(x_t; \sqrt{\bar{\alpha}_t}x_0, (1-\bar{\alpha}_t)I)} \tag{39}$$

$$\propto \exp\left\{-\left[\frac{(x_t - \sqrt{\alpha_t}x_{t-1})^2}{2(1-\alpha_t)} + \frac{(x_{t-1} - \sqrt{\bar{\alpha}_{t-1}}x_0)^2}{2(1-\bar{\alpha}_{t-1})} - \frac{(x_t - \sqrt{\bar{\alpha}_t}x_0)^2}{2(1-\bar{\alpha}_t)}\right]\right\} \tag{40}$$

$$= \exp\left\{-\frac{1}{2}\left[\frac{(x_t - \sqrt{\alpha_t}x_{t-1})^2}{1-\alpha_t} + \frac{(x_{t-1} - \sqrt{\bar{\alpha}_{t-1}}x_0)^2}{1-\bar{\alpha}_{t-1}} - \frac{(x_t - \sqrt{\bar{\alpha}_t}x_0)^2}{1-\bar{\alpha}_t}\right]\right\} \tag{41}$$

$$= \exp\left\{-\frac{1}{2}\left[\frac{-2\sqrt{\alpha_t}x_tx_{t-1} + \alpha_t x_{t-1}^2}{1-\alpha_t} + \frac{x_{t-1}^2 - 2\sqrt{\bar{\alpha}_{t-1}}x_{t-1}x_0}{1-\bar{\alpha}_{t-1}} + C(x_t, x_0)\right]\right\} \tag{42}$$

$$\propto \exp\left\{-\frac{1}{2}\left[\frac{-2\sqrt{\alpha_t}x_tx_{t-1}}{1-\alpha_t} + \frac{\alpha_t x_{t-1}^2}{1-\alpha_t} + \frac{x_{t-1}^2}{1-\bar{\alpha}_{t-1}} - \frac{2\sqrt{\bar{\alpha}_{t-1}}x_{t-1}x_0}{1-\bar{\alpha}_{t-1}}\right]\right\} \tag{43}$$

$$= \exp\left\{-\frac{1}{2}\left[\left(\frac{\alpha_t}{1-\alpha_t} + \frac{1}{1-\bar{\alpha}_{t-1}}\right)x_{t-1}^2 - 2\left(\frac{\sqrt{\alpha_t}x_t}{1-\alpha_t} + \frac{\sqrt{\bar{\alpha}_{t-1}}x_0}{1-\bar{\alpha}_{t-1}}\right)x_{t-1}\right]\right\} \tag{44}$$

Upon further expansion of Eq. 44,

$$= \exp\left\{-\frac{1}{2}\left[\frac{\alpha_t(1-\bar{\alpha}_{t-1})+1-\alpha_t}{(1-\alpha_t)(1-\bar{\alpha}_{t-1})}x_{t-1}^2 - 2\left(\frac{\sqrt{\alpha_t}x_t}{1-\alpha_t}+\frac{\sqrt{\bar{\alpha}_{t-1}}x_0}{1-\bar{\alpha}_{t-1}}\right)x_{t-1}\right]\right\} \tag{45}$$

$$= \exp\left\{-\frac{1}{2}\left[\frac{\alpha_t-\bar{\alpha}_t+1-\alpha_t}{(1-\alpha_t)(1-\bar{\alpha}_{t-1})}x_{t-1}^2 - 2\left(\frac{\sqrt{\alpha_t}x_t}{1-\alpha_t}+\frac{\sqrt{\bar{\alpha}_{t-1}}x_0}{1-\bar{\alpha}_{t-1}}\right)x_{t-1}\right]\right\} \tag{46}$$

$$= \exp\left\{-\frac{1}{2}\left[\frac{1-\bar{\alpha}_t}{(1-\alpha_t)(1-\bar{\alpha}_{t-1})}x_{t-1}^2 - 2\left(\frac{\sqrt{\alpha_t}x_t}{1-\alpha_t}+\frac{\sqrt{\bar{\alpha}_{t-1}}x_0}{1-\bar{\alpha}_{t-1}}\right)x_{t-1}\right]\right\} \tag{47}$$

$$= \exp\left\{-\frac{1}{2}\left(\frac{1-\bar{\alpha}_t}{(1-\alpha_t)(1-\bar{\alpha}_{t-1})}\right)\left[x_{t-1}^2 - 2\left(\frac{\sqrt{\alpha_t}x_t}{1-\alpha_t}+\frac{\sqrt{\bar{\alpha}_{t-1}}x_0}{1-\bar{\alpha}_{t-1}}\right)\frac{(1-\alpha_t)(1-\bar{\alpha}_{t-1})}{1-\bar{\alpha}_t}x_{t-1}\right]\right\} \tag{48}$$

$$= \exp\left\{-\frac{1}{2}\left(\frac{1-\bar{\alpha}_t}{(1-\alpha_t)(1-\bar{\alpha}_{t-1})}\right)\left[x_{t-1}^2 - 2\frac{\sqrt{\alpha_t}(1-\bar{\alpha}_{t-1})x_t + \sqrt{\bar{\alpha}_{t-1}}(1-\alpha_t)x_0}{1-\bar{\alpha}_t}x_{t-1}\right]\right\} \tag{49}$$

$$\propto \mathcal{N}\left(x_{t-1};\frac{\sqrt{\alpha_t}(1-\bar{\alpha}_{t-1})x_t + \sqrt{\bar{\alpha}_{t-1}}(1-\alpha_t)x_0}{1-\bar{\alpha}_t},\frac{(1-\alpha_t)(1-\bar{\alpha}_{t-1})}{1-\bar{\alpha}_t}I\right) \tag{50}$$

From Eq. 50 it can be seen that,

$$\sigma_q^2(t) = \frac{(1-\alpha_t)(1-\bar{\alpha}_{t-1})}{(1-\bar{\alpha}_t)} \qquad \mu_q(t) = \frac{\sqrt{\alpha_t}(1-\bar{\alpha}_{t-1})x_t + \sqrt{\bar{\alpha}_{t-1}}(1-\alpha_t)x_0}{1-\bar{\alpha}_t}$$

Now further substituting $x_0 = \frac{x_t-\sqrt{1-\bar{\alpha}_t}\,\epsilon_0}{\sqrt{\bar{\alpha}_t}}$ from Eq. 37 in the mean term $\mu_q(t)$, we get,

$$\mu_q(t) = \frac{\sqrt{\alpha_t}(1-\bar{\alpha}_{t-1})x_t + \sqrt{\bar{\alpha}_{t-1}}(1-\alpha_t)x_0}{1-\bar{\alpha}_t} \tag{51}$$

$$= \frac{\sqrt{\alpha_t}(1-\bar{\alpha}_{t-1})x_t + \sqrt{\bar{\alpha}_{t-1}}(1-\alpha_t)\left(\frac{x_t-\sqrt{1-\bar{\alpha}_t}\epsilon_0}{\sqrt{\bar{\alpha}_t}}\right)}{1-\bar{\alpha}_t} \tag{52}$$

$$= \frac{\sqrt{\alpha_t}(1-\bar{\alpha}_{t-1})x_t + (1-\alpha_t)\left(\frac{x_t-\sqrt{1-\bar{\alpha}_t}\epsilon_0}{\sqrt{\alpha_t}}\right)}{1-\bar{\alpha}_t} \tag{53}$$

$$= \frac{\sqrt{\alpha_t}(1-\bar{\alpha}_{t-1})x_t}{1-\bar{\alpha}_t} + \frac{(1-\alpha_t)x_t}{(1-\bar{\alpha}_t)\sqrt{\alpha_t}} - \frac{(1-\alpha_t)\sqrt{1-\bar{\alpha}_t}\epsilon_0}{(1-\bar{\alpha}_t)\sqrt{\alpha_t}} \tag{54}$$

$$= \left(\frac{\sqrt{\alpha_t}(1-\bar{\alpha}_{t-1})}{1-\bar{\alpha}_t} + \frac{1-\alpha_t}{(1-\bar{\alpha}_t)\sqrt{\alpha_t}}\right)x_t - \frac{(1-\alpha_t)\sqrt{1-\bar{\alpha}_t}}{(1-\bar{\alpha}_t)\sqrt{\alpha_t}}\epsilon_0 \tag{55}$$

$$= \left(\frac{\alpha_t(1-\bar{\alpha}_{t-1})}{(1-\bar{\alpha}_t)\sqrt{\alpha_t}} + \frac{1-\alpha_t}{(1-\bar{\alpha}_t)\sqrt{\alpha_t}}\right)x_t - \frac{(1-\alpha_t)\sqrt{1-\bar{\alpha}_t}}{\sqrt{\alpha_t}}\epsilon_0 \tag{56}$$

$$= \frac{\alpha_t-\bar{\alpha}_t+1-\alpha_t}{(1-\bar{\alpha}_t)\sqrt{\alpha_t}}x_t - \frac{(1-\alpha_t)\sqrt{1-\bar{\alpha}_t}}{\sqrt{\alpha_t}}\epsilon_0 \tag{57}$$

$$= \frac{1-\bar{\alpha}_t}{(1-\bar{\alpha}_t)\sqrt{\alpha_t}}x_t - \frac{(1-\alpha_t)\sqrt{1-\bar{\alpha}_t}}{\sqrt{\alpha_t}}\epsilon_0 \tag{58}$$

$$= \frac{1}{\sqrt{\alpha_t}}x_t - \frac{(1-\alpha_t)\sqrt{1-\bar{\alpha}_t}}{\sqrt{\alpha_t}}\epsilon_0 \tag{59}$$

Along the similar lines we can prove that, $\mu_\theta(t) = \frac{1}{\sqrt{\alpha_t}}x_t - \frac{(1-\alpha_t)\sqrt{1-\bar{\alpha}_t}}{\sqrt{\alpha_t}}\epsilon_\theta(x_t,t)$

### A.1.3 Proof of Theorem-1

The variational divergence term in Eq. 1 is:

$$D_{KL}\left(Q(\theta)\middle\|Z\cdot\frac{P(\theta|D_f,D_r)}{P(D_f|\theta)}\right) = \mathbb{E}_{Q(\theta)}\left[\ln\frac{Q(\theta)P(D_f|\theta)}{Z\cdot P(\theta|D_f,D_r)}\right] \tag{60}$$

$$\overset{(g)}{=} \mathbb{E}_{Q(\theta)}\left[\ln P(D_f|\theta)\right] + \mathbb{E}_{Q(\theta)}\left[\ln\frac{Q(\theta)}{P(\theta|D_f,D_r)}\right] \tag{61}$$

$$\overset{(h)}{=} \mathbb{E}_{Q(\theta)}\left[\sum_{x_0\in D_f}\ln P(x_0|\theta)\right] + \mathbb{E}_{Q(\theta)}\left[\ln\frac{Q(\theta)}{P(\theta|D_f,D_r)}\right] \tag{62}$$

$$= \underbrace{\mathbb{E}_{\theta\sim Q(\theta)}\left[\sum_{x_0\in D_f}\ln P(x_0|\theta)\right]}_{\text{ⓐ}} + \underbrace{D_{KL}(Q(\theta)||P(\theta|D_f,D_r))}_{\text{ⓑ}} \tag{63}$$

$(g)$ holds as the normalization constant is independent of $\theta$. $(h)$ is true because of the i.i.d. assumption on the data. We further derive the terms ⓐ and ⓑ of Eq. 63 below.

• **Further derivation of term ⓐ:** For further simplification of this term, we would require some more theoretical formulations as follows:

**Lemma 3** *The Kullback-Leibler divergence for two multivariate normal distributions is given by:*

$$D_{KL}(N(x;\mu_x,\Sigma_x)\|N(y;\mu_y,\Sigma_y)) = \frac{1}{2}\left[\log\frac{|\Sigma_y|}{|\Sigma_x|} - d + tr(\Sigma_y^{-1}\Sigma_x) + (\mu_y-\mu_x)^T\Sigma_y^{-1}(\mu_y-\mu_x)\right]$$

**Proof 2** *Proof of this lemma can be found in any standard information theory textbook and is avoided here.*

From Lemma 2, we know that $q(x_{t-1}|x_t,x_0) = \mathcal{N}(x_{t-1};\mu_q(t),\sigma_q^2(t)I)$ with $\mu_q(t) = \frac{1}{\sqrt{\alpha_t}}x_t - \frac{1-\alpha_t}{\sqrt{1-\bar{\alpha}_t}\sqrt{\alpha_t}}\epsilon_0$ and $p_\theta(x_{t-1}|x_t) = \mathcal{N}(x_{t-1};\mu_\theta(t),\sigma_q^2(t)I)$ with $\mu_\theta(t) = \frac{1}{\sqrt{\alpha_t}}x_t - \frac{1-\alpha_t}{\sqrt{1-\bar{\alpha}_t}\sqrt{\alpha_t}}\epsilon_\theta(x_t,t)$ where $\sigma_q^2(t) = \frac{(1-\alpha_t)(1-\bar{\alpha}_{t-1})}{(1-\bar{\alpha}_t)}$. Now using this and the above Lemma 3, the KL-divergence in the ELBO term of Lemma 1 can be written as follows:

$$L_t = D_{KL}(q(x_{t-1}|x_t,x_0)\,\|\,p_\theta(x_{t-1}|x_t)) \tag{64}$$

$$= D_{KL}\left(N(x_{t-1};\mu_q,\Sigma_q(t))\,\|\,N(x_{t-1};\mu_\theta,\Sigma_q(t))\right) \tag{65}$$

$$= \frac{1}{2\sigma_q^2(t)}\left\|\frac{1}{\sqrt{\alpha_t}}x_t - \frac{1-\alpha_t}{\sqrt{1-\bar{\alpha}_t}\sqrt{\alpha_t}}\epsilon_\theta(x_t,t) - \frac{1}{\sqrt{\alpha_t}}x_t + \frac{1-\alpha_t}{\sqrt{1-\bar{\alpha}_t}\sqrt{\alpha_t}}\epsilon_0\right\|_2^2 \tag{66}$$

$$= \frac{1}{2\sigma_q^2(t)}\left\|\frac{1-\alpha_t}{\sqrt{1-\bar{\alpha}_t}\sqrt{\alpha_t}}\epsilon_0 - \frac{1-\alpha_t}{\sqrt{1-\bar{\alpha}_t}\sqrt{\alpha_t}}\epsilon_\theta(x_t,t)\right\|_2^2 \tag{67}$$

$$= \frac{1}{2\sigma_q^2(t)}\left\|\frac{1-\alpha_t}{\sqrt{1-\bar{\alpha}_t}\sqrt{\alpha_t}}(\epsilon_0 - \epsilon_\theta(x_t,t))\right\|_2^2 \tag{68}$$

$$= \frac{(1-\alpha_t)^2}{2\sigma_q^2(t)(1-\bar{\alpha}_t)\alpha_t}\|\epsilon_0 - \epsilon_\theta(x_t,t)\|_2^2 \tag{69}$$

$$= \frac{(1-\alpha_t)}{\alpha_t(1-\bar{\alpha}_{t-1})}\|\epsilon_0 - \epsilon_\theta(x_t,t)\|_2^2 \tag{70}$$

The first term ⓐ of Eq. 63 can now be expressed using Monte Carlo estimation as follows:

$$\text{ⓐ} = \mathop{\mathbb{E}}_{\theta \sim Q(\theta)} \left[ \sum_{x_0 \in D_f} \ln P(x_0|\theta) \right] \tag{71}$$

$$\approx \frac{1}{M} \sum_{s=1}^{M} \left[ \sum_{x_0 \in D_f} \ln P(x_0|\theta^s) \right] \tag{72}$$

$$\overset{(i)}{\gtrsim} \frac{1}{M} \sum_{s=1}^{M} \left[ -\sum_{x_0 \in D_f} \sum_{t=2}^{T} \mathop{\mathbb{E}}_{q(x_t|x_0)} [D_{KL}(q(x_{t-1}|x_t, x_0)||p_{\theta^s}(x_{t-1}|x_t))] \right] \tag{73}$$

$$\overset{(j)}{\approx} -\sum_{x_0 \in D_f} \sum_{t=2}^{T} \mathop{\mathbb{E}}_{q(x_t|x_0)} \left[ \frac{(1-\alpha_t)}{\alpha_t(1-\bar{\alpha}_{t-1})} ||\epsilon_0 - \epsilon_\theta(x_t, t)||^2 \right] \tag{74}$$

$(i)$ holds as a consequence of Lemma 1. Now using a crude estimate of $M = 1$ (i.e. denoting $\theta^1 = \theta$ as we are minimizing with respect to the variable $\theta$) and equation 70, $(j)$ holds.

• **Further Derivation of term ⓑ:** Using the above Lemma 3 with the variational posterior distribution $Q(\theta) = \prod_{i=1}^{d} \mathcal{N}(\theta_i, \sigma_i^2)$ and the posterior distribution with full data $P(\theta|D_r, D_f) = \prod_{i=1}^{d} \mathcal{N}(\mu_i^*, \sigma_i^{*2})$, this term can be simplified as follows:

$$D_{KL}(Q(\theta)||P(\theta|D_f, D_r)) = \sum_{i=1}^{d} \left[ \ln \frac{\sigma_i^*}{\sigma_i} + \frac{\sigma_i^2 + (\theta_i - \mu_i^*)^2}{2\sigma_i^{*2}} - \frac{1}{2} \right] \tag{75}$$

Finally, incorporating both the terms ⓐ and ⓑ terms of Eq. 74 and Eq. 75 respectively we get,

$$D_{KL}\left( Q(\theta) \middle| \middle| Z \cdot \frac{P(\theta|D_f, D_r)}{P(D_f|\theta)} \right) \gtrsim \underbrace{-\sum_{x_0 \in D_f} \sum_{t=2}^{T} \mathop{\mathbb{E}}_{q(x_t|x_0)} \left[ \frac{(1-\alpha_t)}{\alpha_t(1-\bar{\alpha}_{t-1})} ||\epsilon_0 - \epsilon_\theta(x_t, t)||^2 \right]}_{\text{I}}$$

$$+ \underbrace{\sum_{i=1}^{d} \left[ \frac{(\theta_i - \mu_i^*)^2}{2\sigma_i^{*2}} + \frac{\sigma_i^2}{2\sigma_i^{*2}} + \log \frac{\sigma_i^*}{\sigma_i} - \frac{1}{2} \right]}_{\text{II}}$$

#### A.1.4 Loss function Derivation

• **Adaptation of part-$I$ in VDU:** Note that the term $I$ of the lower bound above is computationally expensive and memory intensive because for each $x_0 \in D_f$ we need to add noise for all values of $t \in \{2, \ldots, T\}$ and then denoise again for each $t$. Now, if $T$ is very large, this essentially increases the sample size to $\mathcal{O}(T)$. Thus, to avoid this, if we simply apply the idea from the previous work (Ho et al., 2020), this can be written as follows:

$$I = -\sum_{x_0 \in D_f} \sum_{t=2}^{T} \mathop{\mathbb{E}}_{q(x_t|x_0)} \left[ \frac{(1-\alpha_t)}{\alpha_t(1-\bar{\alpha}_{t-1})} ||\epsilon_0 - \epsilon_\theta(x_t, t)||^2 \right] \tag{76}$$

$$\cong - \mathop{\mathbb{E}}_{\substack{x_0 \sim D_f \\ \epsilon_0, t}} \left[ \frac{(1-\alpha_t)}{\alpha_t(1-\bar{\alpha}_{t-1})} ||\epsilon_0 - \epsilon_\theta(\sqrt{\bar{\alpha}_t}x_0 + \sqrt{1-\bar{\alpha}_t}\epsilon_0, t)||^2 \right] \tag{77}$$

- **Adaptation of part-$II$ in VDU:** Assuming $P(\theta|D_r)$ is mean-shifted version of $P(\theta|D_r, D_f)$ with the same variance factor i.e. with $\sigma_i = \sigma_i^*$, term $II$ turns out to be as follows:

$$II = \sum_{i=1}^{d} \left[ \frac{(\theta_i - \mu_i^*)^2}{2\sigma_i^{*2}} + \frac{\sigma_i^2}{2\sigma_i^{*2}} + \log \frac{\sigma_i^*}{\sigma_i} - \frac{1}{2} \right]$$

$$= \sum_{i=1}^{d} \left[ \frac{(\theta_i - \mu_i^*)^2}{2\sigma_i^{*2}} + \frac{1}{2} + \log 1 - \frac{1}{2} \right]$$

$$= \sum_{i=1}^{d} \left[ \frac{(\theta_i - \mu_i^*)^2}{2\sigma_i^{*2}} \right]$$

This simplified version is used as the second component of our proposed loss function. The reason for assuming $\sigma_i = \sigma_i^*$ is to simplify the optimization process. However, one can remove this assumption and optimize with respect to $\sigma_i$ to find the optimal variance factor of $P(\theta|D_r)$.

## A.2    Additional Results

### A.2.1    Quantitative Results using VDU

Table 4 presents the unlearning performance of the VDU method across various classes from MNIST, CIFAR-10, and tinyImageNet datasets for different values of the hyperparameter $\gamma$. Increasing $\gamma$ can improve unlearning efficacy (higher PUL) in some cases but may degrade sample quality (higher u-FID), and vice versa.

Table 4: PUL($\uparrow$) and u-FID($\downarrow$) for class unlearning on the MNIST, CIFAR-10, and tinyImageNet datasets for different values of $\gamma$.

| Dataset | Unlearned Classes | $\gamma = 0.1$ | | $\gamma = 0.3$ | | $\gamma = 0.6$ | | $\gamma = 0.8$ | |
|---|---|---|---|---|---|---|---|---|---|
| | | PUL(%) | u-FID | PUL(%) | u-FID | PUL(%) | u-FID | PUL(%) | u-FID |
| MNIST | Class-0 | 61.00% | 29.34 | 65.25% | 36.75 | 63.50% | 35.13 | 62.00% | 37.15 |
| | Class-1 | 75.06% | 14.43 | 72.15% | 14.33 | 55.21% | 11.54 | 71.67% | 13.12 |
| | Class-2 | 69.08% | 40.15 | 62.53% | 33.50 | 65.84% | 35.87 | 60.10% | 38.28 |
| | Class-3 | 43.70% | 29.74 | 40.00% | 31.86 | 44.20% | 33.47 | 42.72% | 36.95 |
| | Class-6 | 50.52% | 37.42 | 46.52% | 38.35 | 50.17% | 38.24 | 48.61% | 38.19 |
| | Class-7 | 40.90% | 8.36 | 30.17% | 15.52 | 35.77% | 11.35 | 36.70% | 9.01 |
| | Class-8 | 68.96% | 38.20 | 69.11% | 43.59 | 72.91% | 44.68 | 74.23% | 42.57 |
| | Class-9 | 44.43% | 23.19 | 44.73% | 26.98 | 48.49% | 26.01 | 46.67% | 29.58 |
| CIFAR-10 | Class-0 (airplane) | 53.18% | 28.93 | 48.96% | 31.04 | 46.01% | 30.66 | 39.55% | 32.34 |
| | Class-1 (Automobile) | 60.87% | 30.86 | 56.18% | 33.68 | 57.12% | 28.72 | 68.29% | 34.26 |
| | Class-6 (Frog) | 63.98% | 34.22 | 62.56% | 30.17 | 70.04% | 34.99 | 51.64% | 29.72 |
| | Class-7 (Horse) | 41.81% | 25.19 | 40.45% | 23.40 | 55.93% | 24.09 | 48.36% | 20.17 |
| | Class-8 (Ship) | 71.63% | 24.46 | 74.65% | 28.75 | 56.54% | 20.51 | 69.95% | 19.64 |
| | Class-9 (Truck) | 44.60% | 32.02 | 43.89% | 34.50 | 41.00% | 33.79 | 48.18% | 29.18 |
| tinyImageNet | Class-0 (Fish) | 54.54% | 23.46 | 49.30% | 27.56 | 58.75% | 29.92 | 56.67% | 33.18 |
| | Class-75 (Butcher Shop) | 60.10% | 27.84 | 55.65% | 21.17 | 66.67% | 24.17 | 58.86% | 20.92 |
| | Class-125 (Soda Bottle) | 50.00% | 27.48 | 42.64% | 18.16 | 51.60% | 19.17 | 49.85% | 23.15 |

## A.2.2    Results on CelebA

Table 5 compares our method with the baseline methods on unlearning different features of the CelebA (Liu et al., 2015) dataset. VDU achieves the best u-FID across all features, indicating it produces the highest-quality images post-unlearning. While ESD and SalUn often achieve higher or comparable PUL, they suffer from extremely high u-FID scores, especially for Eyeglasses and Bangs (u-FID > 275), suggesting degraded image quality. VDU balances these metrics, offering significantly better visual fidelity while still achieving competitive PUL.

- **Experimental details for CelebA:** For the pre-trained model, we have used the following open-source architecture implementation: `https://github.com/tqch/ddpm-torch` and trained the model for 40 epochs with a learning rate of $2^{-5}$. For this, $T = 1000$ and batch size is taken to be 128. The pre-trained model has

Table 5: PUL(↑) and u-FID(↓) for features unlearning on the CelebA dataset.

| Dataset | Unlearned Classes | Fine-tuning | | SalUn | | ESD | | VDU (Our Method) | |
|---------|-------------------|-------------|--------|-------|--------|-----|--------|------------------|--------|
| | | PUL (%) | u-FID | PUL (%) | u-FID | PUL (%) | u-FID | PUL (%) | u-FID |
| **CelebA** | Hats | 83.56 | 11.73 | 37.67 | 98.17 | 38.34 | 82.18 | **43.56** | **27.46** |
| | Eyeglasses | 87.18 | 10.36 | 89.39 | 282.63 | **89.80** | 275.01 | 57.67 | **24.79** |
| | Bangs | 85.52 | 12.75 | **67.87** | 310.31 | 67.50 | 299.37 | 46.65 | **23.17** |

FID of 17.82. For the unlearning, we achieve the optimal results using $\gamma = 0.3$ just for 2 epochs of training with our method.

### A.2.3 Class Unlearned Samples using VDU

Visual results of the original images and unlearned images for various class unlearning settings on the MNIST, CIFAR-10, and tinyImageNet datasets are displayed in Figure 7. It can be seen that after unlearning, VDU produces images of high quality.

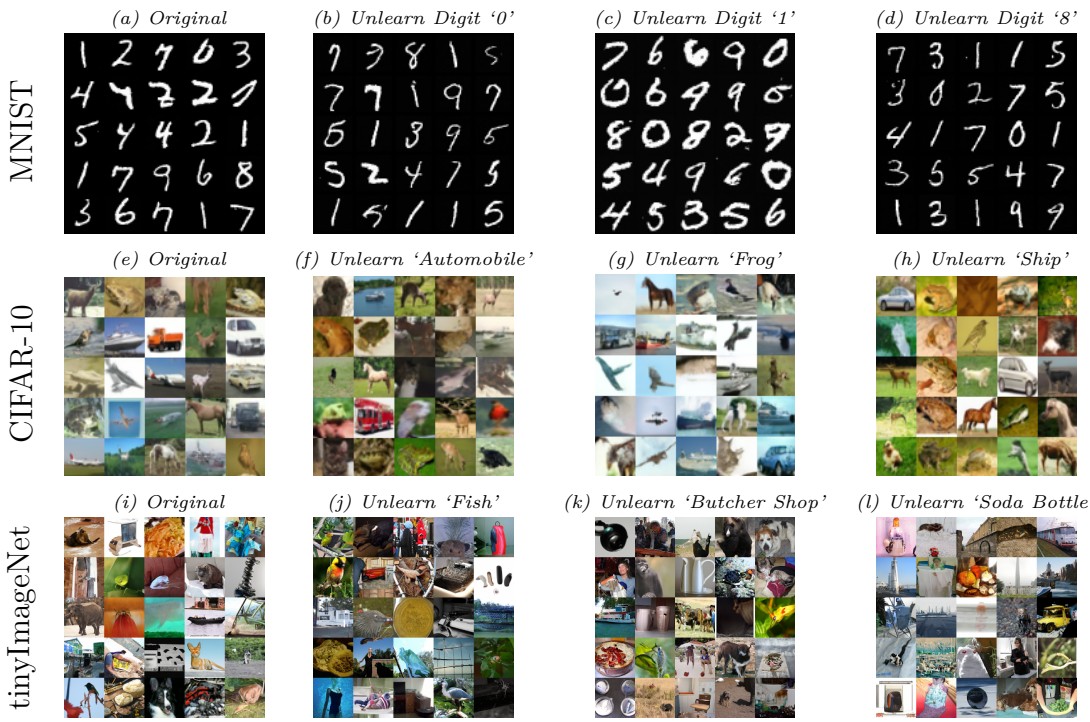

Figure 7: Generated samples from the pre-trained DDPM model and different class unlearned models. Our method shows superior image quality after unlearning.

### A.2.4 Feature Unlearned Samples using VDU

Visual results of the original images and unlearned images over various epochs for various feature unlearning settings, such as 'artistic style' and 'a particular object' from a Stable Diffusion model pre-trained on the LAION-5B (Schuhmann et al., 2022) dataset, are depicted in Figure 8, and Figure 9, respectively. Results of unlearning 'Van Gogh style' from Figure 8 show that given a prompt related to the generation of 'Van Gogh' style images, the unlearned model doesn't generate such artistic style. Similarly, results for unlearning 'car' object are also depicted in Figure 9. Visual results below show VDU's progress over multiple epochs during the unlearning. This kind of object removal has multiple use cases in computer vision-related tasks.

To further evaluate feature unlearning in Stable Diffusion, we compute CLIP similarity scores between the text prompt and the generated image, both before and after unlearning. Effective feature suppression is

Unlearning Concept = "Van_Gogh"

Prompt = "a barn on a quiet farm in the style of van gogh"

| Original | Unlearned epoch=1 | Unlearned epoch=2 | Unlearned epoch=3 | Unlearned epoch=4 | Unlearned epoch=5 |

Unlearning Concept = "Van_Gogh"

Prompt = "a busy street in the style of van gogh"

| Original | Unlearned epoch=1 | Unlearned epoch=2 | Unlearned epoch=3 | Unlearned epoch=4 | Unlearned epoch=5 |

Unlearning Concept = "Van_Gogh"

Prompt = "a farm in the style of van gogh"

| Original | Unlearned epoch=1 | Unlearned epoch=2 | Unlearned epoch=3 | Unlearned epoch=4 | Unlearned epoch=5 |

Unlearning Concept = "Van_Gogh"

Prompt = "a snowy forest in the style of van gogh"

| Original | Unlearned epoch=1 | Unlearned epoch=2 | Unlearned epoch=3 | Unlearned epoch=4 | Unlearned epoch=5 |

Unlearning Concept = "Van_Gogh"

Prompt = "a sunrise in the style of van gogh"

| Original | Unlearned epoch=1 | Unlearned epoch=2 | Unlearned epoch=3 | Unlearned epoch=4 | Unlearned epoch=5 |

Figure 8: Results of VDU on unlearning 'Van Gogh style' over multiple epochs

Unlearning Concept = "Car"

Prompt = "a car going downhill"

Unlearning Concept = "Car"

Prompt = "a car going uphill"

Unlearning Concept = "Car"

Prompt = "a car in the mountains"

Unlearning Concept = "Car"

Prompt = "a car near a tunnel"

Figure 9: Results of VDU on unlearning 'car object' over multiple epochs

Unlearning Concept = "Thomas_Kinkade"

Prompt = "a classic country farmhouse surrounded by sunflowers by thomas kinkade"

Unlearning Concept = "Thomas_Kinkade"

Prompt = "thomas kinkade inspired depiction of a gazebo in a blooming garden"

Figure 10: Results of VDU on unlearning the artistic style of 'Thomas Kinkade' over multiple epochs

expected to result in a reduced CLIP score after unlearning. We use the pretrained CLIP ViT-B/32 model to extract image–text representations and compute the similarity scores. As shown in Table 6, the CLIP scores consistently decrease after unlearning, indicating successful suppression of the targeted features. Here for artistic style we report the clip score after 5th epoch os unlearning and for object after 10 the epoch of unlearning. We also see a consistent decrease of score over the epochs indicating a steady suppression of features as epochs increase.

Table 6: CLIP scores before and after unlearning for different feature types.

| Feature Type | Prompt | Original Image | Unlearned Image |
|---|---|---|---|
| Artistic Style | a barn on a quiet farm in the style of van Gogh | 0.3582 | 0.2593 |
| | a farm in the style of van Gogh | 0.3364 | 0.2482 |
| | a sunset in the style of van Gogh | 0.3477 | 0.2212 |
| Object | a car going downhill | 0.2729 | 0.2132 |
| | a car in the mountains | 0.2976 | 0.2136 |

### A.3   Further Discussions

### A.3.1   Minimization of Lower Bound

Notice that Theorem 1 establishes a lower bound to our variational objective which is minimized in our proposed loss function. In general, there is no theoretical guarantee that minimizing the derived lower bound necessarily minimizes the original objective. However, it is important to note that this lower bound arises exclusively from the application of Lemma 1, which corresponds to the Evidence Lower Bound (ELBO) formulation for diffusion models. In general, the ELBO becomes tight only when the model is sufficiently well trained and can be assumed to accurately represent the underlying data distribution. Under this assumption, minimizing the ELBO serves as a meaningful proxy for minimizing the original objective. Consequently, our experimental goal is to demonstrate that, despite optimizing a lower bound in the loss function, the method still achieves the desired unlearning objective in practice.

### A.3.2   Estimation of $\mu_i^*$ and $\sigma_i^*$

As can be seen from our method's formulation, we require the model parameters' mean ($\mu_i^*$) and variance ($\sigma_i^*$), which can only be obtained from multiple independent training checkpoints. So we train 5 models independently on each of the datasets to calculate $\mu_i^*$ and $\sigma_i^*$ for MNIST, CIFAR-10, and tinyImageNet datasets. Since pre-trained checkpoints $\theta^*$ are essentially samples from $P(\theta|D_r, D_f)$, i.e. $\theta^* \sim P(\theta|D_r, D_f)$, according to the Bayesian inference (Blundell et al., 2015) formulation, these can be estimated from the empirical mean and variance of multiple independent training checkpoints of the pre-trained model. Therefore, by varying the number of checkpoints achieved from independent trainings, Table 7 shows that our approach appears to be robust and effective even with a relatively small number of independent training checkpoints.

Table 7: Impact of the number of checkpoints from multiple independent training runs

| # Checkpoints | MNIST (Digit-1) | | CIFAR-10 (Ship) | | tIN (Butcher Shop) | |
|---|---|---|---|---|---|---|
| | PUL | u-FID | PUL | u-FID | PUL | u-FID |
| 2 | 75.54 | 13.12 | 72.23 | 28.98 | 62.36 | 26.74 |
| 3 | 70.46 | 12.34 | 70.22 | 30.16 | 62.00 | 26.56 |
| 4 | 77.97 | 13.22 | 74.65 | 28.75 | 65.86 | 26.47 |

Table 8: Impact of the number of checkpoints from a single training run

| # Checkpoints | MNIST (Digit-1) | | CIFAR-10 (Ship) | | tIN (Class-75) | |
|---|---|---|---|---|---|---|
| | PUL | u-FID | PUL | u-FID | PUL | u-FID |
| 2 | 73.12 | 12.65 | 70.53 | 32.02 | 58.75 | 32.84 |
| 3 | 66.83 | 10.47 | 65.89 | 33.27 | 60.12 | 26.73 |
| 4 | 73.37 | 12.23 | 72.67 | 29.82 | 63.36 | 27.92 |

- **Necessity regarding multiple independent checkpoints:** We acknowledge that the necessity of checkpoints from multiple independent training to estimate $\mu_i^*$ and $\sigma_i^*$ could be a limiting factor of VDU. To get around this problem, we estimate $\mu_i^*$ and $\sigma_i^*$ by saving several checkpoints from a single training run during the final few iterations. Even though theoretically these checkpoints are not independent samples

from $P(\theta|D_r, D_f)$, practically this seems to work (with slight degradation) as shown in Table 8. Also, we believe that this is more realistic because, in general, we save several checkpoints during a single training run. In fact this is done for all the experiments in Stable Diffusion. As it is practically hard to run such large-scale models independently multiple times, we use multiple checkpoints of the Stable-diffusion model from a single training run. Figure 7 and Figure 8 prove that VDU is effective in more practical scenarios.

In both of these configurations (Table 7 and Table 8), we maintain $\gamma$ at the optimal values, which are $\gamma = 0.1$ for MNIST, $\gamma = 0.3$ for CIFAR-10, and $\gamma = 0.6$ for tinyImageNet.

### A.3.3 Computational and Memory cost

- **VDU's computational and memory cost:** The computational cost and memory usage depend on datasets and models. We used two Nvidia A6000 (48 GB GPU) to run all experiments. On the MNIST dataset, per epoch unlearning takes around 50 secs, on CIFAR-10 210 secs, and tinyImageNet takes around 300 secs per epoch. Now, the main memory usage for VDU comes from saving multiple checkpoints. For MNIST per checkpoint takes memory of 40 MB, for CIFAR-10, 200 MB, for tinyImageNet 450 MB per checkpoint, and for CelebA, it is 560 Mb per checkpoint. For Stable Diffusion, the memory usage of each pre-trained checkpoint is 3.97 GB.

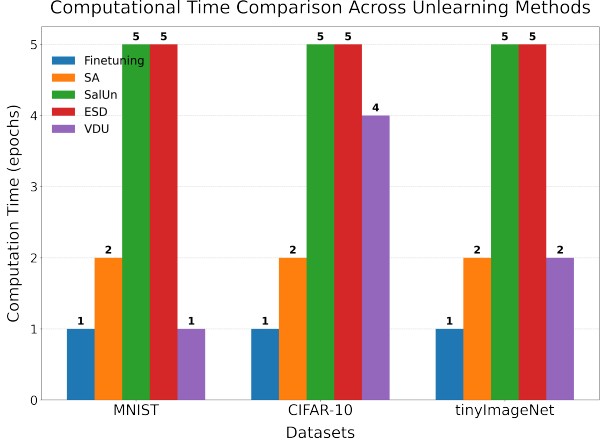

Table 9: Time taken to unlearn for one epoch (in seconds) across datasets.

| Dataset | VDU | SA | SalUN | ESD |
|---|---|---|---|---|
| MNIST | 50 | 540 | 63 | 68 |
| CIFAR-10 | 210 | 588 | 230 | 227 |
| tinyImageNet | 300 | 612 | 489 | 471 |

Figure 11: Unlearning time (epochs) using different methods. VDU takes comparatively fewer epochs than other baselines.

- **Comparison with baselines:** The computing time (in epochs) needed by various unlearning techniques on four datasets—MNIST, CIFAR-10, TinyImageNet, and CelebA—is contrasted in Figure 11. Fine-tuning (blue) has the lowest computational cost but is ineffective when the inaccessible non-unlearning class dataset ($D_r$). In all datasets, VDU (purple bars) consistently uses fewer epochs than ESD (Gandikota et al., 2023b) (red) and SalUn (Fan et al., 2024) (green), and comparable epochs as SA (Heng & Soh, 2023). VDU is an effective unlearning technique that drastically lowers the number of epochs needed without sacrificing performance. This makes it a sensible option for practical uses when computational cost is a crucial consideration.

### A.3.4 Discussion on experimental settings

- **Restriction of access to the samples from $D_r$:** In all of our experimental settings, we don't assume access to samples (original from training data or generated by the pre-trained model) from $D_r$. The primary motivation for this restriction is that we experimentally demonstrate, simply Fine-tuning with samples from $D_r$ (provided we have access to data from $D_r$) is the optimal technique (Ref. third column of Table 1) that produces the best results with the least amount of computing (with only one epoch). Thus, from a practical perspective, all the unlearning methods are only useful when we are restricted to using samples from $D_r$. For this reason, we remove generative replay from Selective Amnesia (SA) method, and the loss components pertaining to $D_r$ is also removed from SalUn and ESD methods.

Table 10: Comparison of unlearning performance with additional fine-tuning using a small subset of $D_r$.

| Dataset | Unlearned Class | Selective Amnesia | | SalUn | | ESD | | VDU ($\gamma = 0.3$) | |
|---------|----------------|--------|-------|--------|-------|--------|-------|--------|-------|
| | | PUL(%) | u-FID | PUL(%) | u-FID | PUL(%) | u-FID | PUL(%) | u-FID |
| MNIST | Digit-1 | 33.64 | 108.76 | 69.87 | 31.67 | 72.86 | 25.38 | 89.08 | 7.46 |
| CIFAR-10 | Automobile | 18.67 | 74.12 | 16.87 | 59.19 | 12.08 | 62.65 | 72.56 | 22.02 |
| tinyImageNet | Fish | 69.83 | 31.86 | 44.59 | 40.92 | 43.87 | 42.15 | 62.96 | 25.82 |

To further analyze performance under limited access to samples from $D_r$, we report additional results in Table 10. In this setting, we fine-tune the unlearned model for 1 epoch using a restricted subset of $D_r$, consisting of only 50 samples per class. The results demonstrate that even limited fine-tuning on $D_r$ consistently improves performance across all methods, and VDU gives relatively superior to comparative performance in this setting also.

- **Restriction of Unlearning epochs:** Note that the unlearning algorithm's main goal is to emulate the retraining method that uses samples only $D_r$ while being computationally efficient. If the unlearning process takes a long time, especially more than retraining, it effectively defeats the purpose of proposing such algorithms. In this case, if we assume finetuning with samples from $D_r$ is feasible, then it is the optimal method both in terms of proposed metrics and computational efficiency (Ref. column 3 in Table 1). Finetuning with $D_r$ achieves optimal results just with 1 epoch (very few iteration depending upon the batch size). We only train all methods for very few epochs ($10\times$ of fine-tuning time) in order to consider computational efficiency and choose the best method that requires the fewest iterations.

### A.3.5 Methodological comparison between SA and VDU

SA method depends on the importance-based penalty using Fisher Information Matrix (FIM), which is avoided in our case with the use of stability regularizer. With the stability regularizer, each parameter is weighted proportionally to $\mathcal{O}(\frac{1}{\sigma^{*2}})$. However, the FIM-based penalty is inherently data-dependent. Specifically, when limited to the unlearning data subset $D_f$, the calculation of the FIM in SA method does not account for the importance of the weights in retaining the information of $D_r$. Consequently, the model forgets the retaining data subset, leading to the generation of low-quality samples (high u-FID scores) even though it performs well in terms of PUL scores.

### A.3.6 Feature entanglement between $D_r$ and $D_f$

In complex datasets due to high entanglement between features, it is really challenging to unlearn a particular specific feature while preserving others. As any machine unlearning approach will unlearn data-level knowledge, the unlearned model will forget some part of the entangled features as well if $D_r$ and $D_f$ overlap. This will have an impact on the unlearning quality (u-FID scores). If there is feature overlap between the samples from $D_r$ and $D_f$, the unlearned model will forget some of the common features because the goal of any machine learning technique is to forget all the data-level information. For example, in feature unlearning from pre-trained Stable Diffusion models, we can visually inspect from Figure 7 and Figure 8 that during the unlearning of artistic styles and objects, image quality is impacted a bit. This emphasizes for possibilities for better methodologies as a part of future work.

### A.4 Implementation Details

### A.4.1 Datasets and Models

- **MNIST:** The MNIST dataset consist of $28\times28$ grayscale representing handwritten digits from 0 to 9. The MNIST dataset contains 60,000 training images and 10,000 testing images. We have used the same architectural model detailed in the open-source implementation: `https://github.com/explainingai-code/DDPM-Pytorch` (Ronneberger et al., 2015) for the MNIST dataset.

- **CIFAR-10:** CIFAR-10 consists of 60,000 color images of size 32 distributed across 10 classes with 6000 images in each class. We adopt an unconditional DDPM model based on the approach by

(Nichol & Dhariwal, 2021), utilizing their official implementation provided in `https://github.com/openai/improved-diffusion`

- **tinyImageNet:** This contains 100000 images of 200 classes (500 for each class) downsized to 64×64 colored images. Each class has 500 training images, 50 validation images and 50 test images. We adopt an unconditional DDPM model based on the approach by (Nichol & Dhariwal, 2021), utilizing their official implementation provided in `https://github.com/openai/improved-diffusion`

- **LAION-5B:** This is the largest public image-text dataset, containing over 5.8 billion samples. More details on this dataset can be found here: `https://laion.ai/projects/`. For this experiment, we use a state-of-the-art Stable Diffusion model. Details of the architecture can be found in the open source implementation at: `https://github.com/CompVis/stable-diffusion`.

### A.4.2   Pre-Training

- **MNIST:** For the pre-training of the DDPM model on MNIST, we adopt the hyperparameters from the open source code base mentioned above. Specifically, a randomly initialized DDPM model is pre-trained on the train set of MNIST and optimized using a learning rate of $10^{-4}$ for 40 epochs with a batch size of 64. The diffusion process follows a linear noise scheduling strategy with $\alpha_1 = 0.0001$ and $\alpha_T = 0.02$ at the final time-step $T$=1000. We have trained 5 models to get the model mean $\mu_i^*$ and variance $\sigma_i^*$ parameter.

- **CIFAR-10:** We use a unconditional DDPM checkpoint, pre-trained on CIFAR-10, from `https://github.com/openai/improved-diffusion`. We fine-tune this model on the train set of CIFAR-10 for 90k iterations, with a batch size of 128, and a learning rate of $10^{-5}$. The total diffusion steps $T = 4000$ with a cosine noise scheduling strategy with $\alpha_1 = 0.0002$ and $\alpha_T = 0.04$. Here, we have trained 4 models to get the model mean $\mu_i^*$ and variance $\sigma_i^*$ parameters. For the pre-training of both DDPM models, we use Adam optimizer.

- **tinyImageNet:** We use a unconditional DDPM checkpoint, pre-trained on tinyImageNet, from `https://github.com/openai/improved-diffusion`. We fine-tune this model on the train set of tinyImageNet for 70k iterations, with a batch size of 64, and a learning rate of $10^{-6}$. The total diffusion steps $T = 4000$ with a cosine noise scheduling strategy with $\alpha_1 = 0.0002$ and $\alpha_T = 0.04$. Here, we have trained 5 models to get the model mean $\mu_i^*$ and variance $\sigma_i^*$ parameters. For the pre-training of both DDPM models, we use the Adam optimizer.

- **LAION-5B:** For this, we didn't train the Stable Diffusion models but used the open-source checkpoints available at: `https://github.com/CompVis/stable-diffusion`. These checkpoints are trained with a linear noise scheduling strategy from $\alpha_1 = 0.00085$ to $\alpha_T = 0.0120$ with $T = 1000$ and learning rate $= 10^{-4}$. We have used four checkpoints that are present in the repository: sd-v1-1.ckpt, sd-v1-2.ckpt, sd-v1-3.ckpt, sd-v1-4.ckpt.

### A.4.3   Unlearning

After pre-training the DDPM models on their respective datasets, as described earlier, we now outline the unlearning process for each. We optimize using our proposed loss function, as defined in Eq. 2, ensuring effective feature removal while maintaining model performance. Note that during the unlearning phase, for all the datasets, the noise scheduler is kept the same as in the pre-training phase.

1. **MNIST:** The model is optimized for unlearning over only 1 epoch using the Adam optimizer with a learning rate of $10^{-6}$ and a batch size of 128.

2. **CIFAR-10:** The model is optimized for unlearning over 4 epochs using the Adam optimizer with a learning rate of $10^{-6}$ and a batch size of 128.

3. **tinyImageNet:** The model is optimized for unlearning over only 2 epochs using the Adam optimizer with a learning rate of $10^{-6}$ and a batch size of 128.

4. **LAION-5B:** Note that, as we don't have access to the labels in this dataset to obtain $D_f$, we generate samples from $D_f$ by prompting ChatGPT and passing these prompts to a model initialized with checkpoint sd-v1-1.ckpt. We only generated 650 samples corresponding to $D_f$ for both 'artistic style' and 'object' unlearning settings. Here we have used Adam optimizer with a learning rate of $10^{-6}$.

### A.4.4   Additional Details

- **Hyper-parameter details:** For all of the benchmark datasets, we select and report the results for $\gamma \in$ {0.1, 0.3, 0.6, 0.8}. We achieve optimal experimental results using $\gamma$=0.1 and 0.6 for all the classes on MNIST and tinyImageNet, respectively. On CIFAR-10, we choose optimal $\gamma$ of 0.1 for unlearning "Automobile" and "Ship". For "Frog", we use $\gamma$=0.3. For feature unlearning in the LAION-5B dataset, we have set $\gamma = 0.6$.

- **Classifier Details:** To calculate u-FID and PUL metrics from unconditional Diffusion models, we need samples that correspond to $D_r$ part of the training data. Thus, we classify samples that relate to $D_r$ by pre-trained models. We trained a ResNet-50 model with an accuracy of 99% for MNIST, a DenseNet-121 with an accuracy of 92% for CIFAR-10, ViT-B16 with an accuracy of 98% for tinyImageNet, and ResNet-50 with an accuracy of 81% for the CelebA dataset.

- **Code:** Our implementation can be found at: `https://github.com/Subhodip123/VDU`. Keep referring to this repository as we keep updating with more implementation details and results.

