# OpenReview forum: "Unlearning in Diffusion models under Data Constraints: A Variational Inference Approach"
_TMLR — Accepted by TMLR_

### Review · Reviewer_igS7 · 2025-12-15

**Summary Of Contributions:**

This paper is on unlearning in the context of (image) diffusion models in the setting where the original pertaining dataset is not accessible.
Their method, based on variational inference, attempts to optimize an objective consisting of two terms -- The first term (plasticity inducer) reduces the likelihood of the undesirable points, and the second term (stability regularizer) works to maintain the quality of generations.
This is in contrast to several current methods which instead either require access to the entire training set, or rely on synthesizing fake data to retrain the model.

The the paper shows:
1. A theoretical guarantee demonstrating that under the gaussian mean-field approximation in parameter space, the ELBO may be lower bounded by the appropriate objective consisting of the plasticity inducer and stability regularizer.
2. They propose a novel algorithm (VDU) that optimizes the above objective to find a model which has forgotten the desired information.
2. Evidence on various image datasets of unlearning.

**Audience:**

Yes

**Audience Explanation:**

This is a paper on diffusion models and unlearning, and will be of relatively high interest to the TMLR audience. The current evidence is only strong for some kind of coarse disentanglement.

**Claims And Evidence:**

Yes

**Claims Explanation:**

There are 5 claims made in the paper:

1. Mathematical derivation of the claim that unlearning can be modeled as a VI problem. They demonstrate this with proof in the Gaussian mean-field approximation setting

2. VDU does not need the original dataset to forget the information. They provide experiments on CIFAR-10 and CelebA-HQ. They measure the FID on the retain class, and this remains low, wheras baseline show a spike in FID when retain data is missing.

3. VDU provides convergence in one-epoch and has superior efficiency. The paper plots "Forgetting score" vs "training steps" and these show that the forgetting metric plateaus early in the training process.

4. VDU achieves disentangled learning. The authors claim that their method can erase a specific style or object without damaging closely related concepts. *evidence for this is a little weaker than the rest*, because the specific classes like Van Gogh vs Church etc seem fairly distinct. It would be nicer if they provided evidence of the concepts being very related and still remaining unaffected (i.e. the paper fails to provide strong evidence for highly entangled concepts).

**Requested Changes:**

It would be good to provide more evidence for the effect of VDU on entangled concepts (such as wheels and cars)

---

> ### Author Response · Authors · 2026-01-10
> **Rebuttal by Authors**
>
> We thank the reviewer for the positive comments about the paper. Specific doubts and changes are clarified below.
>
> **More evidence for the effect of VDU on entangled concepts**
> - Thanks for the suggestion. Due to time constraints, we are unable to find very entangled concepts from the LAION-5B dataset. To provide further evidence of unlearning, we have tried to unlearn another artistic concept. These results are added in Figure 10 of Appendix Section A.2.4.
> - Note that artistic style is an entangled concept because it is related to many concepts, such as different objects, background, shapes etc. that are present in the images. For more evidence, we urge the reviewer to look at Figure 8 and Figure 9 of Appendix Section A.2.4. Similarly, Figure 9 provides more evidence of the unlearning of the car object, which is also an entangled feature.
> - We would like to highlight that we notice in complex datasets that unlearning a highly entangled feature generally degrades the quality of the other features. This is one of the limitation which we tackle as a part of future work. We have already added a detailed discussion on this in Appendix Section A.3.7.

---

### Review · Reviewer_Grhj · 2025-12-20

**Summary Of Contributions:**

This paper proposes Variational Diffusion Unlearning (VDU), a data efficient method for machine unlearning in diffusion models (DM) built upon the Variational Bayesian Unlearning framework in [1]. More precisely, similarly to [1], VDU minimizes the variational KL-divergence $D_{\text{KL}}(q(\theta)|p(\theta|D_r))$ between the learnable approximate posterior $q(\theta)$ and the true posterior $p(\theta|D_r)$ defined via the Bayes’ theorem. Here $D_r$ denotes the retain set. As in [1], this KL-divergence is represented as a sum of two terms: the log likelihood of the forget set $D_f$ and the KL-divergence between the learnable approximate posterior belief $q(\theta)$ and the fixed approximate posterior belief $p(\theta|D_r, D_f)$.

The authors propose to model both $q(\theta) \sim \prod_i \mathcal{N}(\theta_i, \sigma_i^{\*2})$  and $p(\theta|D_r, D_f) \sim \prod_i \mathcal{N}(\mu_i^{\*},\sigma_i^{\*2})$ as factorised Gaussians with the shared variance $\sigma_i^{\*2}$. Parameters $\mu_i^{\*}$ and $\sigma_i^{\*2}$ are estimated by computing statistics of either the final checkpoints of $N$ independent runs, or $N$ late checkpoints of the same run.

Finally, the authors derive a lower bound on their likelihood term, which results in the lower bound on the initial variational KL-divergence $D_{\text{KL}}(q(\theta)|p(\theta|D_r))$ minimized w.r.t. $\theta$.

VDU is evaluated in the class unlearning setting on MNIST, CIFAR-10 and tinyImageNet using unconditional DDPM, and in the feature unlearning setting on LAION-5B using Stable Diffusion. In the setting where $D_r$ is unavailable, the proposed method outperforms Selective Amnesia (SA), Saliency Unlearning (SalUn) and Erasing Stable Diffusion (ESD).

**Strengths**:
- For the most part, the paper is well-written, and individual sections are easy to follow.
- Data efficiency: relying only on the forget set $D_f$ might be beneficial when there is no access to the retain set $D_r$.
- VDU outperforms the considered baselines in terms of Percentage of Unlearning (PUL, measures unlearning quality) and Unlearned Fréchet Inception Distance (u-FID, measures utility preservation).

**Weaknesses**:
- [Major] The claim about “the establishment of a theoretical connection between the problem of machine unlearning and variational inference” being a key contribution is somewhat misleading since representation of the problem as approximate Bayesian unlearning has been explored before, notably in [1], which the authors acknowledge to be inspired by (see Sec. 3.2).
- [Major] The authors propose to minimize the *lower* bound on the variational KL-divergence $D_{\text{KL}}(q(\theta)|p(\theta|D_r))$. Lower bound minimization does not guarantee that the function of interest is actually minimized unless you can bound the gap between the lower bound and this function. This weakens the claim about “the established theoretical connection” even more.
- [Major] The logic of model presentation is hard to follow. For full derivation of the final loss in Eq. (2), the reader has to go back and forth between different following sections in the main paper and in the appendix, sometimes not even in the order of appearance. This negatively affects the readability of the paper.
- [Major] Several benchmarks for DM unlearning are missing, such as ConceptBench [2], UnlearnCanvas [3] or I2P.
- [Major] Qualitative results on Celeb-A are missing. Qualitative results for all baselines and all benchmarks are also missing. This makes it impossible to comprehensively evaluate the performance of different approaches.
- [Minor] Qualitative results of object unlearning for Stable Diffusion in Figures 4 and 9 are not convincing and demonstrate significant degradation in the quality of generated images.
- [Minor] The proposed method lacks flexibility and does not scale straightforwardly when the retain set $D_r$ is available, at least partially.

[1] Nguyen Q.P. et al. “Variational Bayesian Unlearning”. In NeurIPS, 2020.

[2] Zhang E. et al. “Forget-Me-Not: Learning to Forget in Text-to-Image Diffusion Models”. In arXiv preprint arXiv:2211.08332, 2023.

[3] Zhang Y. et al. “UnlearnCanvas: Stylized Image Dataset for Enhanced Machine Unlearning Evaluation in Diffusion Models”. In NeurIPS, 2024.

**Audience:**

Yes

**Audience Explanation:**

The problem of machine unlearning has been attracting a lot of attention recently, motivated by a number of use cases including privacy regulation or content moderation. The findings of this paper can potentially be of use to the community of researchers interested in machine unlearning for DMs in data restricted scenarios when only the “to-be-removed” data is available.

**Claims And Evidence:**

Yes

**Claims Explanation:**

Overall, the claims made in the submission are supported reasonably well. The paper indeed proposes a data-efficient method for DM unlearning that requires access only to the forget set $D_f$, and this method is evaluated in the class-unlearning setup on DDPM and in the feature-unlearning setup on Stable Diffusion. However, according to the authors, the key contribution of the paper is supposed to be “the establishment of a theoretical connection between the problem of machine unlearning and variational inference”, which is a misleading claim for two reasons: 1) the same approximate Bayesian unlearning framework has already been proposed in prior works (as mentioned above), and 2) the resulting objective function does not guarantee minimization of the function for which this theoretical connection has been established.

**Requested Changes:**

**Major issues**:
- I would recommend rewriting the claims regarding the theoretical connection between machine unlearning and variational inference as a major contribution, especially since derivation of the variational KL-divergence is not new [1], and since the final loss in Eq. (2) does not guarantee to minimise the desired objective (variational KL-divergence). Alternatively, the authors could provide theoretical proofs that such a guarantee exists.
- The readability of the paper could be significantly improved by changing the order of sections and, as a results, the order of equations. In the current version, the general formulation of the problem in Eq. (1) is immediately followed by the final version of the loss in Eq. (2), without proper annotation of variables and/or parameters which are introduced much later. The order of derivation, however, seems to be as follows: Eq.(1) -> Eq. (51-51) in Sec. A.1.3 -> Lemmas 1 and 2 in Sec. 3.3 -> proofs in Sec. A.1.1. and Sec. A.1.2.3, etc. Some equations (e.g., Eq. (51-54) in Sec. A.1.3) could be moved to the main paper for more seamless flow of information.
- To strengthen the claims about data efficiency, it would be interesting to see comparison between VDU and SA, SalUn and ESD when the latter have access to a small subset of $D_r$ (e.g., to a random subset). Reusing only a small fraction of $D_r$ can still be considered *data efficient* and should not be ignored.
- On each benchmark, it would be interesting to see qualitative comparison with all the baselines to better understand how well u-FID metric captures degradation in image quality.
- Quantitative results lack comparison with the “gold” model, i.e., the model that has been trained from scratch only on $D_r$. Even though this baseline is computationally inefficient, I think the community would benefit from seeing the “upper bound” on the considered metrics.
- The paper would benefit from a more comprehensive evaluation on recent benchmarks (ConceptBench [2], UnlearnCanvas [3], I2P).



**Minor issues**:
- The purpose of Algorithm 1 is not clear, as it describes the standard SGD update. It could be removed, and the free space could be used for equations and/or sections from the appendix that are more critical for understanding the paper.
- A number of equations are not numerated (both in the main paper and in the appendix). Adding numeration would simplify referencing and discussion about the content of the paper.
- It is not clear how the learned approximate posterior belief $q(\theta)$ is used in practice: do you use the mean $\theta$ in $q(\theta) \sim \mathcal{N}(\theta, \sigma^{*2})$ as the parameters of the unlearned model or do you sample from this distribution?
- It would be interesting to see how sensitive are VDU and the baselines to the number of unlearning epochs. This parameter seems to be important for VDU since the first term in Eq. (2) is unbounded, so I wonder if eventually it overpowers the second term.
- In Sec. 4.3, why is there a need to use a classifier “to remove unlearning data from the real data”? Usually D_f is provided, so obtaining D_r is trivial. Maybe the authors meant something else, this should be clarified.
- The order of the columns (metrics) in Tables 1 and 3 is different. The readability would improve if the order becomes the same.
- The reference “Zoubin Ghahramani and H. Attias. Online variational bayesian learning. *Workshop on Online Learning*, NeurIPS, 2020.” does not seem to exist (to the best of my knowledge, there were no workshops with such a title at NeurIPS in 2020). Please, verify that the references are correct.


[1] Nguyen Q.P. et al. “Variational Bayesian Unlearning”. In NeurIPS, 2020.

[2] Zhang E. et al. “Forget-Me-Not: Learning to Forget in Text-to-Image Diffusion Models”. In arXiv preprint arXiv:2211.08332, 2023.

[3] Zhang Y. et al. “UnlearnCanvas: Stylized Image Dataset for Enhanced Machine Unlearning Evaluation in Diffusion Models”. In NeurIPS, 2024.

---

> ### Author Response · Authors · 2026-01-10
> **Rebuttal by Authors - Major Concerns**
>
> We thank the reviewer for the constructive feedback about the paper. We clarify all the doubts and changes in 2 part rebuttal due to character constraints.
>
> **1. Claims regarding the theoretical connection**
> - We appriciate reviewer's concern and pointer to the relevant literature. We have rewritten our claims regarding this contribution.
> - In general, there is no theoretical guarantee that minimizing the derived lower bound necessarily minimizes the original objective. However, it is important to note that this lower bound arises exclusively from the application of Lemma 1, which corresponds to the Evidence Lower Bound (ELBO) formulation for diffusion models. In general, the ELBO becomes tight only when the model is sufficiently well trained and can be assumed to accurately represent the underlying data distribution. Under this assumption, minimizing the ELBO serves as a meaningful proxy for minimizing the original objective. Consequently, our experimental goal is to demonstrate that, despite optimizing a lower bound in the loss function, the method still achieves the desired unlearning objective in practice. We discuss this in detail in Appendix Section A.3.1.
>
> **2. Re-ording of the Sections**
> - We apologize for the trouble. We have tried to reorder the sections for better readability. We added the preliminary on Diffusion models before introducing the variational divergence formulation. We hope that this will help the transition from equation 1 to equation 2 now.
> - Further, according to your suggestion for the better readablity we have added Equations 54-57 in the main paper, which may help with the better flow.
>
> **3. Compare with limited access to $D_r$**
> - The below table gives result of finetuning using $D_r$. We take only 50 samples per class from $D_r$ and finetune the unlearned model on this subset of $D_r$ for 1 epoch. The results demonstrate that even limited fine-tuning on $D_r$ consistently improves performance across all methods, and VDU gives relatively superior to comparative performance in this setting also. We have added these results in Table-10 of Appendix A.3.5.
>
> | Dataset | Unlearned Class | Selective Amnesia PUL (%) | Selective Amnesia u-FID | SalUn PUL (%) | SalUn u-FID | ESD PUL (%) | ESD u-FID | VDU (γ=0.3) PUL (%) | VDU (γ=0.3) u-FID |
> |-|-|-|-|-|-|-|-|-|-|
> | MNIST | Digit-1 | 33.64 | 108.76 | 69.87  | 31.67 | 72.86 | 25.38 | 89.08 | 7.46 |
> | CIFAR-10 | Automobile | 18.67   | 74.12  | 16.87 | 59.19 | 12.08 | 62.65 | 72.56| 22.02 |
> | tinyImageNet | Fish | 69.83  | 31.86 | 44.59 | 40.92 | 43.87 | 42.15 | 62.96 | 25.82 |
>
> **4. Comparisons with retraining on $D_r$**
> - We thank the reviewer for this suggestion. Since retraining from scratch is computationally expensive, we have already included fine-tuning as a strong gold-standard baseline. In this setting, fine-tuning refers to updating the original model using samples from the retained dataset $D_r$. We kindly refer the reviewer to Table 1.
>
> **5. More Benchmarks on CenceptBench and Unlearn Canvas**
> - We thank the reviewer for this suggestion. Due to time and computational constraints, we are unable to provide these results for both benchmarks as these are large scale experimetns. We have therefore identified this direction as part of our future work.
> - However, we would like to point out that ConceptBench uses samples from the LAION dataset to forget three primary categories of concepts: identity, style, and object. Our feature unlearning setup closely follows this setting. Specifically, we use a Stable Diffusion model trained on the LAION-5B dataset to unlearn artistic styles (see Fig. 8) and object categories such as cars (see Fig. 9). In addition, we include Fig. 10 to demonstrate unlearning of an additional artistic style.

---

> ### Author Response · Authors · 2026-01-10
> **Rebuttal by Authors - Minor Concerns**
>
> **1. Remove Algorithm 1**
> - We think that Algorithm 1 clarifies the purpose of what to implement exactly, given the specific input and output. As per the reviewer's suggestion, we have reduced the space for the algorithm and added equations from the appendix to further guide the reader.
>
> **2. Enumeration of Equations**
> - We have tried to enumerate all the equations and reference them. We noticed that the equations in Lemma 2 were not enumerated; we have fixed this.
>
> **3. How do we use $q(\theta)$**
> - Note that we do not explicitly sample from the learned posterior $Q(\theta)$. In the proposed framework, the minimization of variational divergence yields the loss for unlearning. The minimization will cause the transition $P(\theta|D) \to Q(\theta)\approx P(\theta|D_r)$. Thus, one can think that this minimization will result in parameters that are essentially samples from $Q(\theta)$.
>
> **4. Performance of unlearning epochs of VDU**
> - The below table gives the performance of VDU over the epochs. It is noticed that as epochs increases the first term dominates, thus effectively increasing unlearning. However, the quality of the image degrades. We have added these results in Appendix Section A.3.4.
>
> | Dataset        | Unlearned Class | Epoch 1 PUL | Epoch 1 u-FID | Epoch 2 PUL | Epoch 2 u-FID | Epoch 3 PUL | Epoch 3 u-FID | Epoch 4 PUL | Epoch 4 u-FID | Epoch 5 PUL | Epoch 5 u-FID |
> |---------------|----------------|-------------|---------------|-------------|---------------|-------------|---------------|-------------|---------------|-------------|---------------|
> | MNIST         | Digit-1        | 75.06%      | 14.43         | 86.58%      | 33.67         | 97.86%      | 56.17         | 99.55%      | 106.18        | 100.00%     | 218.83        |
> | CIFAR-10      | Automobile     | 27.33%      | 26.82         | 43.67%      | 30.19         | 50.58%      | 29.82         | 60.87%      | 30.86         | 68.97%      | 56.82         |
> | tinyImageNet  | Class-0        | 33.26%      | 18.72         | 58.75%      | 29.92         | 59.86%      | 37.68         | 59.88%      | 52.72         | 60.65%      | 70.95         |
>
> - Note that we already provide variation over epochs for the feature unlearning in the Stable Diffusion model. A similar trend is also visible here.
>
> **5. Why do you need a classifier?**
> - We thank the reviewer for pointing this out. There is a typo in the specific line. We have corrected the specific sentence.
> - The classifier is needed mainly in an unconditional DDPM setting. After unlearning, in order to calculate PUL and u-FID, we need to distinguish the unlearned samples vs. the non-unlearned samples. For example, after unlearning class-1 from MNIST, we need to calculate how many samples belong to non-unlearning classes. Thus classifier is used.
>
> **6. Change the order of Table 3 PUL and u-FID metrics**
> - We apologize for the inconvenience. We have changed the order in Table 3 to be consistent with Table 1.
>
> **7. Reference  Online variational Bayesian learning**
> - We thank the reviewer for pointing this out. We feel there is a typo/mis-reference that has happened.
> - We have removed this reference because we are unable to find it explicitly. However, we would also like to point out that this exact reference also exists in some relevant literature. (Please refer last reference on page 10 in https://arxiv.org/pdf/1710.10628 )

---

### Review · Reviewer_7yLg · 2025-12-28

**Summary Of Contributions:**

This paper addresses the challenge of machine unlearning in diffusion models under data-constrained settings where the full training dataset is inaccessible, proposing Variational Diffusion Unlearning (VDU), which leverages a variational inference framework with a loss function consisting of a plasticity inducer (minimizing log-likelihood of undesired data) and a stability regularizer (preserving pre-trained parameters); it is validated through comprehensive experiments on class unlearning (MNIST, CIFAR-10, tinyImageNet) and feature unlearning (Stable Diffusion), achieving high PUL and low u-FID, while being computationally efficient, requiring only access to the undesired data subset, and effectively balancing unlearning effectiveness and generation quality.

**Audience:**

Yes

**Audience Explanation:**

This study aims to restrict the model from learning certain content, which holds significant value for practical applications by preventing the model from generating unhealthy material.

**Claims And Evidence:**

Yes

**Claims Explanation:**

VDU mathematically formulates unlearning as variational divergence minimization, with a loss function that combines a plasticity inducer (minimizing undesired data log-likelihood) and a stability regularizer, derived from the KL-divergence between the variational posterior and the pre-trained parameters.  Experimentally, VDU achieves high PUL and low u-FID across class unlearning and feature unlearning, outperforming baselines.

**Requested Changes:**

Quantitative Metrics for Feature Unlearning in Stable Diffusion: While Figure 1 qualitatively shows "Van Gogh style" unlearning over epochs, the paper lacks quantitative metrics to measure feature suppression (e.g., style similarity scores using pre-trained style extractors like VGG or CLIP, or text-image alignment metrics for "Car object" unlearning). Without numerical benchmarks, it is unclear if residual style/feature traces remain. The authors could supplement qualitative results with quantitative metrics (e.g., style cosine similarity between original and unlearned outputs) to validate feature unlearning effectiveness.

---

> ### Author Response · Authors · 2026-01-10
> **Rebuttal by Authors**
>
> We thank the reviewer for the positive comments about the paper. Specific doubts and changes are clarified below.
>
> **Quantitative Metrics for Feature Unlearning**
>
> We use the pretrained CLIP ViT-B/32 model to extract image–text representations and compute the similarity scores before and after unlearning. The table below shows that the CLIP scores consistently decrease after unlearning, indicating suppression of the targeted features. We have added these results in Table-5 with a detailed discussion in Appendix section A.2.4.
>
> | Feature Type        | Prompt                                         | CLIP Score (Original Image) | CLIP Score (Unlearned Image) |
> |---------------------|-----------------|-----------------------------|---------------------|
> | Artistic Style      | a barn on a quiet farm in the style of van gogh | 0.3582                      | 0.2593                        |
> | Artistic Style      | a farm in the style of van gogh                 | 0.3364                      | 0.2482                        |
> | Artistic Style      | a sunset in the style of van gogh               | 0.3477                      | 0.2212                        |
> | Object              | a car going downhill                            | 0.2729                      | 0.2132                        |
> | Object              | a car in the mountains                          | 0.2976                      | 0.2136                        |

---

### Author Response · Authors · 2026-01-27
**Regarding the final decision**

Dear Editor,

We are writing to you to enquire about the status of the paper.

It has been a few weeks since we submitted our responses.

It would be very helpful if the decision be conveyed sooner so that we can plan the next steps for the work.

Thank you for your kind consideration.

Authors

Authors

---

> ### Comment · Action_Editor_C8ku · 2026-02-05
>
> Dear Authors,
>
> The final decision will be finalized in the next few days.
>
> Best,
>
> The AE

---

### Decision · Action_Editor_C8ku · 2026-02-16

**Recommendation:** Accept with minor revision

**Additional Comments:**

Please include the two further remarks raised by reviewer Grhj, namely:

- Argument on the choice for Eq. 1 and 2 also in the main paper and not only in the supplementary material.

- Highlight in the limitations that some comparisons with respect with more standard unlearning benchmarks is missing.

**Audience:**

Yes

**Audience Explanation:**

Generation and unlearning are two themes that currently cover great importance in the Machine Learning and Deep Learning community - this work is potentially of broad interest for the community.

**Claims And Evidence:**

Yes

**Claims Explanation:**

This work proposes a method to safely regulate the outputs of diffusion models, which can sometimes generate harmful, violent, or obscene content. While existing machine unlearning approaches attempt to remove such undesirable behaviors by retraining models to forget specific training data, they often require access to the full original dataset, which may not be available in practice. To address this limitation, it is here proposed Variational Diffusion Unlearning (VDU), a computationally efficient method that operates in data-constrained settings and only requires a subset of undesirable training samples. Through variational inference, VDU minimizes a loss function composed of two components: a plasticity inducer that reduces the likelihood of generating outputs related to unwanted data, and a stability regularizer that preserves overall image generation quality by constraining parameter changes. Experimental results on both class unlearning and feature unlearning tasks qualitatively showcase that VDU can suppress undesired content in pre-trained diffusion models while maintaining generation performance.

Although this work has some limitations that should be further put in evidence (as highlighted by reviewer Grhj), within the tested framework the work shows innovation and hints satisfactorily at his functioning.

---

> ### Author Response · Authors · 2026-03-12
> **Changes in accordance to the action editor comments**
>
> We thank all the reviewers and the action editor for their valuable time and feedback. We appreciate all the positive comments about the paper. We hope we were able to clarify the majority of the doubts and make changes to the final manuscripts to the best of our abilities. The majority of changes suggested by the reviewers were included in the previous version. The final camera-ready version includes the action points mentioned by the action editor, which are clarified below.
>
> 1. Note that, following the remarks by Grhj, we have already included the arguments for the choice of equations 1 and 2. We have included an outline of the derivation from equation 1 to equation 2 in the main paper as well. This is highlighted in the proof of Theorem 1 (Refer to equations 8-11).
>
> 2. We have also included an additional limitation that our method needs further empirical evaluation based on the recent unlearning benchmarks. Refer to the third line in the Limitations and Future Works section.